# ATTACK-RESISTANT WATERMARKING FOR AIGC IMAGE FORENSICS VIA DIFFUSION-BASED SEMANTIC DEFLECTION

**Qingyu Liu**[1], **Yitao Zhang**[1], **Zhongjie Ba**[1,2]*, **Chao Shuai**[1],
**Peng Cheng**[1,2], **Tianhang Zheng**[1,2], **Zhibo Wang**[1,2]
[1]The State Key Laboratory of Blockchain and Data Security, Zhejiang University
[2]Hangzhou High-Tech Zone (Binjiang) Institute of Blockchain and Data Security, China
{qingyuliu,12521031,zhongjieba,chaoshuai,peng_cheng,zthzheng,
zhibowang}@zju.edu.cn

## ABSTRACT

Protecting the copyright of user-generated AI images is an emerging challenge as AIGC becomes pervasive in creative workflows. Existing watermarking methods (1) remain vulnerable to real-world adversarial threats, often forced to trade off between defenses against spoofing and removal attacks; and (2) cannot support semantic-level tamper localization. We introduce PAI, a training-free inherent watermarking framework for AIGC copyright protection, plug-and-play with diffusion-based AIGC services. PAI simultaneously provides three key functionalities: robust ownership verification, attack detection, and semantic-level tampering localization. Unlike existing inherent watermark methods that only embed watermarks at noise initialization of diffusion models, we design a novel key-conditioned deflection mechanism that subtly steers the denoising trajectory according to the user key. Such trajectory-level coupling further strengthens the semantic entanglement of identity and content, thereby further enhancing robustness against real-world threats. Moreover, we also provide a theoretical analysis proving that only the valid key can pass verification. Experiments across 12 attack methods show that PAI achieves 98.43% verification accuracy, improving over SOTA methods by 37.25% on average, and retains strong tampering localization performance even against advanced AIGC edits. Our code is available at https://github.com/QingyuLiu/PAI.

## 1 INTRODUCTION

Recent advances in diffusion-based AIGC systems have enabled creators to generate high-quality images at scale, raising urgent demands for copyright protection and provenance accountability (U.S. Copyright Office, 29-Jan-2025). While most prior works on AIGC copyright have focused on protecting training data, we highlight the emerging challenge of safeguarding the copyright of user-generated AIGC images. To this end, watermarking provides a natural solution, where existing approaches can be grouped into embedded and inherent paradigms. Embedded watermarking injects signals after generation through encoder–decoder networks or frequency transforms (Zhang et al., 2024; Fernandez et al., 2023; Zhu et al., 2018), but requires costly adversarial training to improve robustness against common image degradations such as compression. In contrast, inherent watermarking integrates watermarks into the generative process (Wen et al., 2023; Yang et al., 2024b), semantically coupling identity with content and achieving robustness without extra training, making it a more practical direction for protecting user copyrights in real-world AIGC services.

However, beyond common degradations, real-world adversaries exploit stronger threats that compromise copyright protection directly. The removal attack (Saberi et al., 2023; Müller et al., 2024; Ba et al.) erases ownership evidence, the spoofing attack (Yang et al., 2024a; Müller et al., 2024; Dong et al.; Ba et al.) forges false attribution, and the localized tampering attack maliciously alters image semantics while preserving an authentic appearance (e.g., face swapping (Chen et al., 2020)).

---

*The corresponding author.

Collectively, these threats lead to disputes where authorship may be denied, falsely claimed, or obscured by partial manipulation, highlighting the need for watermarking systems that enable both reliable verification and forensic analysis.

When confronted with these escalating threats, existing watermarking schemes face two critical limitations. **First, we observe a critical trade-off between defending against removal and spoofing attacks.** Current schemes rely on one-dimensional verification metrics such as decoded bit counts (Zhu et al., 2018; Tancik et al., 2020; Fang et al., 2022) to decide ownership. This way inevitably collapses the complexity of adversarial behaviors into a single scalar. Removal attacks decrease the score to fall below the threshold, while spoofing inflates it beyond the threshold. Adjusting this threshold for one attack often undermines defense against the other, since defenders cannot assume prior knowledge of the adversarial goal. **Second, most existing methods remain limited to binary ownership verification and lack provenance-aware forensic capabilities.** In practice, defenders must determine not only **whether an image is watermarked** but also **whether and where it has been manipulated**. Ideally, a watermarking system should distinguish between removal (attacked but still owned), spoofing (forged and not owned), and tampering (owned but locally modified). With the rise of advanced AIGC editing tools (Rombach et al., 2022; Google, 2024), which enable subtle semantic-level modifications while preserving global appearance, pixel-based localization methods (e.g., EditGuard (Zhang et al., 2024)) become ineffective. This calls for a more robust forensic analysis that can reason about semantic manipulations in the latent space.

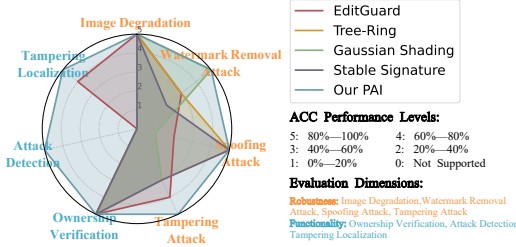

Figure 1: Comparison of SOTA image watermarking methods across 7 dimensions of robustness and functionality. Overall, PAI achieves balanced and superior results across all dimensions.

In this paper, we propose PAI (**P**rovenance-**A**ware **I**nherent watermarking), a plug-and-play training-free framework for AIGC copyright forensics. PAI aligns watermark injection with the two-stage generative process of diffusion: in the initialization stage, a user key and timestamp are embedded into the initial noise through the Box-Muller transformation (Box & Muller, 1958) that preserves the Gaussian prior, ensuring both compatibility with diffusion sampling and diversity of outputs; in the denoising stage, we apply a key-conditioned deflection during early steps, semantically binding the user identity to the generative trajectory. This dual-stage design yields implicit, content-adaptive watermarks that inherently resist inversion-based attacks, as any reversal of the process requires the correct key to reconstruct the initial state.

For watermark provenance, PAI inverts the injection process via DDIM inversion and defines the initialization bias (the discrepancy between theoretical and recovered initial noise) as a unified forensic signal. This enables three tasks: (1) vanilla verification, where theoretical and empirical analysis show that valid keys consistently yield lower bias than invalid or non-watermarked cases, allowing principled rejection via hypothesis testing; (2) robust verification, where instead of scalar scores, we model bias in a low-dimensional latent space to distinguish removal and spoofing through their opposite directional deviations; and (3) semantic tampering localization, where spatial anomalies in bias align with manipulated regions in RGB space, enabling precise semantic-level forensics under PS, Deepfakes, or advanced AIGC-based editing tools.

We evaluate PAI on CelebA-HQ, COCO, and DDPM across degradations and 12 attacks, comparing with SOTA watermark schemes. PAI achieves up to 99.0% detection accuracy under removal, 96.3% under spoofing, and 100% under tampering, with an AUC of 89.77 for localization, remaining effective even against full-image edits from Gemini 2.0 Flash. Moreover, it also withstands black- and white-box adaptive attacks. To sum up, our contributions are:

- We propose PAI, a training-free inherent watermarking scheme for AIGC copyright protection. By combining initialization embedding with a key-conditioned deflection mechanism, our PAI couples the watermark with the generative trajectory, achieving improved robustness.
- We provide a theoretical guarantee on the key exclusivity of PAI, which only allows the owner's key to pass the verification, excluding the possibility of any forged key being accepted.

- We overcome the critical trade-off of existing watermarking methods between defending against removal and spoofing attacks. Extensive evaluations show that our PAI achieves strong robustness against common degradations and 12 attacks, with 98.43% average ownership verification accuracy, outperforming the existing watermark methods by 37.25% on average.
- Our PAI enables reliable localization against semantic-level manipulation by AIGC tools (e.g., Gemini 2.0 Flash), a new threat that bypasses existing pixel-based localization schemes but is effectively addressed by our noise-space localization.

## 2 RELATED WORK

Digital watermarking has long been used for copyright protection and authentication (Xuehua, 2010; Samuel & Penzhorn, 2004), but traditional frequency-domain or cryptographic schemes are vulnerable to modern deep learning–based removal attacks (Saberi et al., 2023; Zhao et al., 2024; Jiang et al., 2023). Recent research therefore focuses on two paradigms: **Embedded watermarking**, which injects signals after generation via encoder–decoder networks (Zhu et al., 2018; Tancik et al., 2020; Jia et al., 2021; Fang et al., 2022), often extending classical DCT/DWT methods (Lan et al., 2023; Jing et al., 2021). While they can jointly optimize imperceptibility and robustness, these schemes usually depend on adversarial training, introduce pixel-level artifacts (e.g., in flat backgrounds), and incur high computational overhead (Huang et al., 2024; Xu et al., 2025). In contrast, **Inherent watermarking** directly integrates watermark signals into the generative process, typically by perturbing diffusion noise representations (Wen et al., 2023; Yang et al., 2024b). This semantic coupling avoids separate encoder–decoder architectures, yields stronger security by making watermarks inseparable from content, offers greater robustness to post-processing, and eliminates training overhead through lightweight, generator-level injection.

**Challenges and motivation.** Despite advances in embedded and inherent watermarking, existing methods show key limitations in AIGC copyright protection, especially under adversarial or forensic scenarios. Functionally, they are mostly confined to binary ownership verification and cannot support richer forensic tasks such as attack detection or tamper localization. For robustness, existing methods remain vulnerable to advanced real-world threats, especially exhibiting a trade-off between removal and spoofing attacks (see Sec. 6.3). This motivates our key insight, consistent with prior observations (Zhao et al., 2024), that watermark robustness tends to increase with stronger semantic coupling between watermark signals and generated content. Therefore, unlike prior inherent methods that inject watermarks only into the initialization noise, we further introduce a key-conditioned semantic deflection (see Sec. 4), which strengthens the coupling between watermark signals and content semantics and thereby improves robustness. In addition, our design also supports attack detection and semantic-level tamper localization, enabling fine-grained forensics beyond binary verification.

## 3 PROBLEM FORMULATION

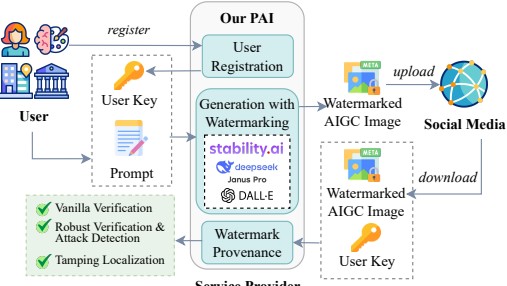

Figure 2: System model of our PAI.

**System Model.** The AIGC copyright protection system involves two entities: users and service providers (shown in Fig. 2). Users generate images via AIGC services and embed watermarks to protect their copyright, while providers may watermark all outputs to ensure provenance and regulatory compliance. To this end, we propose PAI, a plug-and-play and training-free watermarking scheme deployed on the provider side. **Since most current AIGC services are built on diffusion models, PAI is fully compatible with the diffusion generation framework**, eliminating the need for extra encoders/decoders. PAI operates in three phases:

- **User Registration.** Users first register with the provider, which issues a unique private key $K$ as the sole credential for watermarking verification.
- **Generation with Watermarking.** Users access the provider with their private key $K$ and a prompt to generate watermarked AIGC images. Here, the generated image metadata stores a

timestamp-based salt designed for increasing diversity (see Sec.4), which is also required for watermark verification; modifying this timestamp does not affect copyright verification (see Sec. 6.5).

- **Watermark Provenance.** In a copyright dispute, the owner submits the watermarked image and $K$ for watermark provenance. The service provider verifies ownership using the image, its metadata timestamp, and $K$, and can trace the originating user for compliance. Our scheme also detects whether the image has been subjected to removal, forgery, or tampering attacks while preserving correct ownership; for localized tampering, our method can provide the binary mask for localizing the tampered regions.

**Threat Model.** In this paper, we consider the service provider as the defender. We assume that the defender (i.e., the service provider) is trusted, meaning it does not leak user keys and operates over a secure communication channel. Conversely, adversaries can be anyone who wants to modify the watermark of legitimate users' images or exploit the generation service to create poisoned images. We consider a strong adversary who can harvest many watermarked/non-watermarked images, acquire a surrogate diffusion model, and may even obtain white-box access to our models, but cannot compromise the key store. Specifically, we categorize attacks into five escalating types, including degradation, watermark removal, spoofing, localized tampering, and adaptive attacks, and instantiate 16 concrete attacks (detailed in Tab. 4). The attack scenarios are illustrated in Fig. 9.

**Design Goal.** We propose a watermarking scheme that (1) preserve the visual quality of the original AIGC images generated by the provider, (2) enables high-confidence ownership verification tied to each user's private key, (3) resists real-world degradations and adversarial threats including removal, spoofing, localized tampering, and adaptive attacks, and (4) supports precise tamper localization against unauthorized manipulated editing.

## 4 GENERATION WITH WATERMARK

**Preliminary.** Diffusion models serve as the backbone of modern AIGC systems, enabling widely adopted platforms, e.g., Stable Diffusion (Rombach et al., 2022), DALL·E (Ramesh et al., 2022), and MidJourney (Midjourney, 2023). Among them, DDIM (Song et al., 2020) reformulates stochastic diffusion into a deterministic non-Markovian process governed by an ODE, enabling efficient sampling. Given an image, DDIM inversion attempts to project it back into the noise space. The two processes can be written as:

$$
\begin{aligned}
\text{Sampling process: } & x_{t-1} = \frac{1}{\sqrt{\alpha_t}} x_t + \left( \sqrt{1 - \bar{\alpha}_{t-1}} - \frac{1}{\sqrt{\alpha_t}} \sqrt{1 - \bar{\alpha}_t} \right) \varepsilon_\theta \left( x_t, t \right), \\
\text{Inversion process: } & \hat{x}_t = \sqrt{\alpha_t} \hat{x}_{t-1} + \left( \sqrt{1 - \bar{\alpha}_t} - \sqrt{\alpha_t - \bar{\alpha}_t} \right) \varepsilon_\theta \left( \hat{x}_{t-1}, t \right).
\end{aligned}
\tag{1}
$$

In practice, the deviation between inversion and forward sampling processes accumulates as an intrinsic bias, which serves as a key signal for our watermark verification.

**Our Method.** For the generation process of diffusion models, a Gaussian noise $x_t \sim \mathcal{N}(0, 1)$ randomly sampled from the standard normal distribution is initialized as the starting point, and the corresponding image is generated by stepwise denoising (see Eq. 1). Existing inherent watermarking methods embed watermark information directly on $x_t$ to obtain $x_t^{wm}$, followed by standard DDIM sampling to produce watermarked images. In contrast, our watermark injection operates through dual mechanisms: initialization-stage embedding and deflection-stage enhancement during the sampling process. The entire pipeline of our watermark scheme is shown in Fig. 3.

*Initialization Stage.* To bind the generated content with the user identity, we embed a user-specific secret key $K$ into the initialization process. To further enhance output diversity under deterministic samplers, a timestamp-based salt $S$ is incorporated. To preserve theoretical consistency with the standard diffusion framework, we employ the Box–Muller transformation (Box & Muller, 1958), which maps uniform variables to Gaussian ones, thereby integrating $K$ and $S$ into the watermarked initialization. Formally, we define the initialization function $\mathcal{F}$ as:

$$
x_t^{wm} = \mathcal{F}(K, S) = \sqrt{-2 \ln S} \cdot \cos(2\pi \cdot \Phi(K)),
\tag{2}
$$

where $K \sim \mathcal{N}(0, 1)$, $S \sim \mathcal{U}(0, 1)$, and the resulting $x_t^{wm} \sim \mathcal{N}(0, 1)$. The timestamp-based salt $S$ is sampled with the generation time as a seed and stored in the image metadata. The cumulative distribution function $\Phi(\cdot)$ converts $K$ into $\Phi(K) \sim \mathcal{U}(0, 1)$. Since $\Phi(K)$ and $S$ are independent and uniformly distributed, the Box-Muller transformation ensures $x_t^{wm}$ adheres to $\mathcal{N}(0, 1)$.

*Deflection Phase.* To enhance the semantic correlation between watermarks and generated images, we propose progressive watermark injection during sampling. Specifically, we replace the predicted $\hat{x}_0$ with a deflection function $\mathcal{H}\left(K, x_t^{wm}, t\right)$:

$$x_{t-1}^{wm} = \sqrt{\bar{\alpha}_{t-1}} \cdot \mathcal{H}\left(K, x_t^{wm}, t\right) + \sqrt{1 - \bar{\alpha}_{t-1}} \cdot \epsilon_\theta\left(x_t^{wm}, t\right),$$

$$\text{where } \mathcal{H}\left(K, x_t^{wm}, t\right) = \underbrace{(\gamma \cdot K + 1)}_{\text{deflection coefficient}} \cdot \underbrace{\left(\frac{x_t^{wm} - \sqrt{1 - \bar{\alpha}_t}\epsilon_\theta\left(x_t^{wm}, t\right)}{\sqrt{\bar{\alpha}_t}}\right)}_{\text{predicted } \hat{x}_0}, \tag{3}$$

Here, $\hat{x}_0$ denotes the predicted target image. At each step $t$, $\mathcal{H}(\hat{x}_0, K)$ introduces minor deflections guided by $K$, which iteratively accumulate to form semantically coherent watermarks. The hyper-parameter $\gamma$ controls deflection intensity. To balance image quality and robustness (see experiments in Sec. 6), we set $t = 50$ and apply deflection during the first five steps with $\gamma = 0.1$.

**Why Do We Design in This Way?** Our watermark is designed to ensure verifiability, diversity, and robustness. Verifiability is achieved by incorporating a user-specific key $K$ for reliable ownership authentication. Diversity is provided by a timestamp-based salt $S$, which introduces stochasticity to $x_t^{wm}$ and prevents deterministic outputs under fixed prompts. Robustness is strengthened through a key-conditioned deflection that couples the identity signal with the generative trajectory, ensuring that any key mismatch produces structured deviations during inversion. This coupling makes PAI resistant to key extraction and advanced semantic attacks.

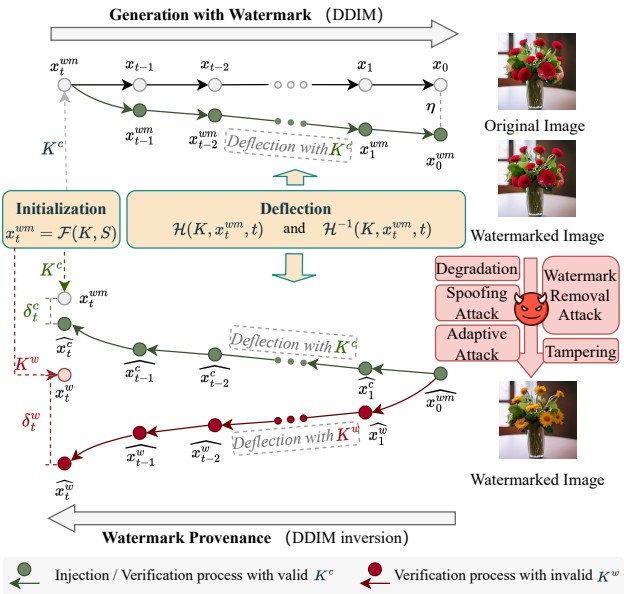

Figure 3: The pipeline of our PAI. The initialization stage aims to calculate the initial noise $x_t$ for generation and verification processes. The deflection stage aims to use the user key $K$ to deflect the generative trajectory. $\eta$ is our implicit content-adaptive watermark.

# 5 WATERMARK PROVENANCE

The generated watermarked image $x_0^{wm}$, when disseminated through social media platforms, becomes vulnerable to adversarial manipulations and unauthorized alterations. Given a watermarked image to be verified $\hat{x}_0^{wm}$, our proposed PAI enables three critical forensic functionalities: (1) vanilla verification to reject non-watermarked content or invalid keys; (2) robust verification to detect attacks and verify ownership under various attacks; and (3) tamper localization to predict manipulated regions.

## 5.1 VANILLA VERIFICATION

The copyright verification process for a given watermarked image aims to determine the image copyright by validating if the user key can successfully authenticate ownership. Our verification methodology is the inversion process of the watermarked image generation process (described in Sec. 4), eliminating the need for additional decoders to extract the hidden messages. Specifically, given a user key $K$, the verification process first maps the watermarked image into the noise space via DDIM inversion process ($\hat{x}_0^{wm} \to \hat{x}_t^{wm}$), which uses the invertible deflection function $\mathcal{H}^{-1}$:

$$\hat{x}_t^{wm} = \sqrt{\bar{\alpha}_t} \cdot \mathcal{H}^{-1}(K, x_{t-1}^{wm}, t) + \sqrt{1 - \bar{\alpha}_t} \cdot \epsilon_\theta\left(x_{t-1}^{wm}, t\right),$$

$$\text{where } \mathcal{H}^{-1}\left(K, x_{t-1}^{wm}, t\right) = \frac{x_{t-1}^{wm} - \sqrt{1 - \bar{\alpha}_{t-1}}\epsilon_\theta\left(x_{t-1}^{wm}, t\right)}{\sqrt{\bar{\alpha}_{t-1}} \cdot (\gamma \cdot K + 1)}. \tag{4}$$

Following this DDIM inversion procedure, we obtain the noise-space representation $\hat{x}_t^{wm}$. Subsequently, the initialization function $\mathcal{F}$ (detailed in Eq. 2) is employed to reconstruct the original initialization point $x_t^{wm}$ generated during watermark embedding. We identify the discrepancy between the initialization $x_t^{wm}$ and the inverted noise representation $\hat{x}_t^{wm}$ as the initialization bias $\delta_t$. The vanilla ownership determination is decided by a threshold $\tau_{vanilla}$:

$$\mathbb{E}\left[|\delta_t|^2\right] = \left|\hat{x}_t^{wm} - \mathcal{F}(K, S)\right|^2 < \tau_{vanilla}, \tag{5}$$

where $\mathbb{E}\left[|\delta_t|^2\right]$ is the second-order moment of initialization bias.

**Theoretical Analysis.** Due to the imperfect reversibility of DDIM inversion, the recovered noise $\hat{x}_t$ inevitably deviates from the true $x_t$, yielding a non-vanishing intrinsic bias $\delta_t$ (see the above preliminary). Therefore, a critical security question arises: **How do we ensure only the valid key passes verification?** For the theoretical analysis, we consider a generator without modeling error, so that deviations observed in DDIM inversion originate only from intrinsic bias. Consider the worst case where a forged key $K^w$ approaches the valid key $K^c$. We now reformulate our security requirement into a formal statement and prove it as follows.To address this, we provide both a theoretical guarantee under ideal assumptions and empirical evidence under practical conditions.

*Ideal Condition.* We begin our analysis by assuming an ideal generative model where the noise predictor used in the diffusion process is perfectly trained. That is, it accurately predicts the expected noise conditioned on the current latent state at each timestep. This assumption ensures that deviations observed during DDIM inversion arise solely from the intrinsic approximation error of the inversion process itself, not from modeling noise or learning inaccuracies. Under this setting, we examine a worst-case adversarial scenario: the attacker uses a forged key $K^w$ that is arbitrarily close to the valid key $K^c$. This represents the hardest possible case for verification, as the only difference between the two trajectories lies in the key. If we can establish that invalid keys still result in higher bias even in this proximity limit, it implies that verification will succeed with even greater reliability in more practical scenarios. We now reformulate our security requirement into a formal robustness statement and prove it as follows.

*Theorem.* For an invalid key $K^w$ asymptotically approaching the valid key $K^c$, the second-order moment of the initialization bias $\delta_t^c$ (associated with $K^c$) remains smaller than that of $\delta_t^w$ (associated with $K^w$):

$$\lim_{K^w \to K^c} \mathbb{E}\left[|\delta_t^c|^2\right] < \mathbb{E}\left[|\delta_t^w|^2\right] \tag{6}$$

*Proof.* Given verification image $\hat{x}_0^{wm}$, we analyze two distinct inversion trajectories: The authenticated path $\hat{x}_0^c = \hat{x}_t^c$ with valid key $K^c$ yields bias $\delta_t^c = \hat{x}_t^c - x_t^{wm}$, while the unauthenticated path $\hat{x}_0^w \to \hat{x}_t^w$ with $K^w \neq K^c$ produces $\delta_t^w = \hat{x}_t^w - x^w$, where $x^w = \mathcal{F}(K^w, S) \neq x_t^{wm}$. Through systematic derivation (detailed in Appendix A.2), we establish the expressions of the initialization bias:

$$\delta_t^c = \hat{x}_t^c - x_t^{wm} = \sum_{i=1}^t \frac{\sqrt{\bar{\alpha}_t}}{\sqrt{\bar{\alpha}_i} M^{t-i}} F_i^c,$$

$$\delta_t^w = \hat{x}_t^w - x^w = \frac{M^t}{N^t} x_t^{wm} - x^w + \sum_{i=1}^t \frac{\sqrt{\bar{\alpha}_t}}{\sqrt{\bar{\alpha}_i} N^{t-i}} \cdot F_i^w,$$

$$\text{where } F_i^c = \left(\sqrt{1-\bar{\alpha}_i} - \frac{\sqrt{\alpha_i - \bar{\alpha}_i}}{M}\right) \cdot \left(\epsilon_\theta\left(\hat{x}_{i-1}^c, i\right) - \epsilon_\theta\left(x_i^{wm}, i\right)\right),$$

$$\text{and } F_i^w = \left(\sqrt{1-\bar{\alpha}_i} - \frac{\sqrt{\alpha_i - \bar{\alpha}_i}}{N}\right) \varepsilon_\theta\left(\hat{x}_{i-1}^w, i\right) - \frac{M^i}{N^i}\left(\sqrt{1-\bar{\alpha}_i} - \frac{\sqrt{\alpha_i - \bar{\alpha}_i}}{M}\right) \varepsilon_\theta\left(x_i^{wm}, i\right)$$

(7)

where $M = \gamma \cdot K^c + 1$ and $N = \gamma \cdot K^w + 1$. This expression reflects that valid keys only accumulate temporal errors during the inversion process, while invalid keys additionally introduce a baseline discrepancy from mismatched initialization. This ensures their bias cannot be smaller than that of the valid key, even in the worst-case proximity. Complete proof is provided in Appendix A.3.

*Practical Condition.* While the above analysis assumes an idealized generative setting, real-world systems are subject to imperfect training and model noise. While such model imperfections introduce additional noise into the inversion process, their effects are largely shared between valid

and invalid keys through the accumulation of prediction errors in the $F_i$ terms of Eq. 7. However, invalid keys additionally incur a baseline initialization discrepancy, which remains unaffected by model quality. As a result, the bias induced by invalid keys remains consistently larger than that of valid keys, even in the presence of model imperfections. We empirically verify that the theoretical guarantee remains effective in practice through two evaluations: 1) Distribution Visualization. We visualize the second-order moment of the initialization bias induced by the watermarking key. As shown in Fig. 4, invalid keys lead to significantly higher initialization bias compared to the valid key. Furthermore, when applying the verification process to unwatermarked images, the resulting bias is also significantly high. These results support our security analysis by empirically demonstrating that only the correct key generates a statistically small initialization bias. 2) Adaptive Key Extraction Attack. We simulate a white-box attacker with full access to the generation model. The attacker performs gradient-based optimization on a forged key to minimize $|\delta_t|$ with respect to a target image. However, as shown in Sec. 6.5, the optimization fails to reach the bias level of the true key, and the final result remains above the verification threshold. This confirms that even with complete system access, the key cannot be extracted or imitated.

**Vanilla Verification Threshold Decision.** We analyze the distribution of the initialization bias $\delta_t$, as shown in Fig. 4. We observe that the second-order moment of $\delta_t$ remains consistently small for the valid key, but is significantly larger for invalid keys and non-watermarked images. Therefore, assuming the invalid-key bias follows a Gaussian distribution, we formulate vanilla verification as a one-sided hypothesis test. A significance level $\alpha$ determines the threshold $\tau_{vanilla}$, which controls the false positive rate for unauthorized or non-watermarked inputs. If the observed initialization bias exceeds $\tau_{vanilla}$, the key is rejected as invalid; otherwise, $\delta_t$ is further projected into

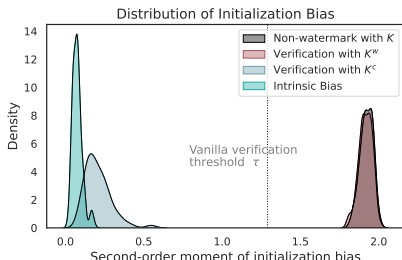

Figure 4: Visualization of the second-order moment of the initialization bias.

the PCA space for robust verification. Importantly, this PCA-based verification specifically targets adversarial cases where attacks force the bias to appear deceptively close to the benign distribution.

## 5.2 ROBUST VERIFICATION

**Observation.** Beyond rejecting invalid keys and non-watermarked images, a robust verifier must also distinguish adversarial threats, especially watermark removal and spoofing, which have opposite targets. Most prior work focuses on matching embedded messages during injection and extraction by using a one-dimensional criterion. Our experiments show this scalar cannot separate removal from spoof-

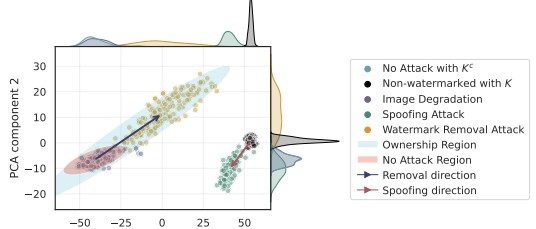

Figure 5: Visualization of the initialization bias under attacks.

ing (see Sec. 6.3). Although both attacks introduce semantic distortions, they may result in similar bias magnitudes, making it difficult to separate them using a single numerical threshold. We observe that, despite similar magnitudes, removal and spoofing induce distinct directions in the high-dimensional latent space. Projecting initialization-bias vectors with PCA reveals separable trajectories (Fig. 5): removal departs from the benign watermarked cluster, whereas spoofing departs from non-watermarked images in the opposite direction.

In the PCA-projected space, we model benign initialization biases as $\mathbf{z} \sim \mathcal{N}(\boldsymbol{\mu}, \boldsymbol{\Sigma})$ and measure deviations with the Mahalanobis distance (Mahalanobis, 2018). Under the benign null, $D^2(\mathbf{z}) \sim \chi_k^2$ with $k$ equal to the projection dimension ($k = 2$ here), giving a threshold $\tau_{robust} = \chi_2^2(1 - \alpha)$. A sample is marked as abnormal if $D^2(\mathbf{z}) > \tau_{robust}$, providing a principled statistical test for detecting adversarial perturbations or watermark inconsistency. To support both attack detection and ownership verification, we define the benign distribution differently for each task. For attack detection, we model the benign distribution from valid-key inversions without attack. For ownership verification, the benign distribution also includes degraded or removal-affected samples, since authorship is determined by the valid key rather than post-modifications. This distinction ensures robustness to removal while preserving legitimate ownership.

### 5.3 TAMPER LOCALIZATION

**Observation.** We observe that regional differences in RGB space consistently align with anomalies in the corresponding noise space (Fig. 6(a),(b)). Specifically, given a valid key and a tampered image $x_0^{wm'}$ ($x_0^{wm'} = \hat{x}_0^{c'}$), we first apply our inverse deflection process ($\hat{x}_0^{c'} \to \hat{x}_t^{c'}$) to obtain its representation in the noise space. The original watermarked image $x_0^{wm} = \hat{x}_0^c$ follows the same process $\hat{x}_0^c \to \hat{x}_t^c$. Therefore, the tampered region in the RGB space is the difference between the tampered and original watermarked image, denoted as $\Omega_0 = x_0^{wm'} - x_0^{wm}$. Extending this to the noise space, we obtain the tampering discrepancy at time step $t$ as $\Omega_t = \hat{x}_t^{c'} - \hat{x}_t^c$. Fig. 6(a) and (b) visualize $\Omega_0$ and $\Omega_t$, showing that anomalous regions in the RGB space correspond to coherent anomalies in the noise space.

While $\Omega_t$ offers accurate tamper localization, it relies on access to the original watermarked image to compute $\hat{x}_t^c$, which is typically unavailable in practice. To address this, we reformulate $\Omega_t = (\hat{x}_t^{c'} - x_t^{wm}) - (\hat{x}_t^c - x_t^{wm}) = \delta_t^{c'} - \delta_t^c$, where $\delta_t^{c'}$ and $\delta_t^c$ represent the initialization biases of the tampered and original watermarked images, respectively. We approximate $\delta_t^c$ with the average intrinsic bias $\bar{\Delta}_t$ from non-watermarked samples, yielding $\hat{\Omega}_t = \delta_t^{c'} - \bar{\Delta}_t$. Experiments on both pixel- and latent-space diffusion models (using 100 clean images) show that $\hat{\Omega}_t$ closely matches $\Omega_t$, effectively exposing tampered regions (Fig. 6(c)).

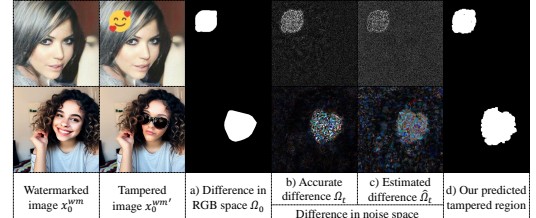

| Watermarked image $x_0^{wm}$ | Tampered image $x_0^{wm'}$ | a) Difference in RGB space $\Omega_0$ | b) Accurate difference $\Omega_t$ | c) Estimated difference $\hat{\Omega}_t$ | d) Our predicted tampered region |
|---|---|---|---|---|---|
| | | | Difference in noise space | | |

Figure 6: Visualization of the difference in noise space between the original and the tampered watermarked images. The anomalies in the tampered region show consistency in RGB space and noise space. The first row is the result of DDPM and the second row is the result of Stable Diffusion.

Finally, traditional image processing techniques (such as filtering and morphological operations) are applied to refine $\hat{\Omega}_t$ into a binary mask that accurately highlights manipulated areas (Fig. 6(d)).

## 6 EVALUATION

In this section, we evaluate PAI against existing watermarking baselines in terms of effectiveness, robustness, tamper localization, and resilience to adaptive attacks. Additional settings (Appendix A.6.1), results, and analyses, including ablation studies (Appendix A.6.3), runtime analysis (Appendix A.6.6), and discussions on the key space (Appendix A.5) are provided.

### 6.1 SETUP

**Implementation Details.** We evaluate our framework in both unconditional and T2I generation settings. For unconditional generation, we adopt a pre-trained DDPM on CelebA-HQ, and for T2I, we use Stable Diffusion v2.1 with captions from COCO and CelebA-HQ. For fairness, each method produces 5,000 watermarked images and shares the same 5,000 non-watermarked images. The complete implementation details are provided in the Appendix A.6.1.

**Evaluation Metrics.** We evaluate our scheme across four aspects: functionality, image quality, robustness, and tamper localization. For verification and detection, we report TPR/FPR and ACC, including attack detection accuracy (A-ACC) and ownership verification accuracy (O-ACC). For perceptual quality, we use CLIP-based similarity (CLIP-Image and CLIP-Prompt), while PSNR is reported under adversarial settings. For tamper localization, we adopt standard segmentation metrics including F1, AUC, and IoU. Detailed definitions are provided in the Appendix A.6.1.

**Comparative Baseline (Defender).** We compare against two embedded methods (Edit-Guard (Zhang et al., 2024), Stable Signature (Fernandez et al., 2023)) and two inherent methods (Tree-Ring (Wen et al., 2023), Gaussian Shading (Yang et al., 2024b)). Among them, only Edit-Guard supports tampering localization.

**Attacker Setting.** We evaluate robustness with four degradations and 12 attacks, including removal, spoofing, localized tampering, and adaptive attacks (see Sec. 3). Detailed attack configurations and hyperparameters are provided in Appendix A.4.2.

### 6.2 EFFECTIVENESS

Tab. 1 presents a comprehensive comparison of our watermarking scheme with existing methods in terms of reliability and fidelity under clean conditions. For reliability, our variants achieve superior

Table 1: Evaluation of watermarking methods under clean settings (no attack). We report detection performance with and without watermarks, as well as image generation quality. Bold values indicate the best results.

| Method | COCO Dataset | | | | | | CelebA-HQ Dataset | | | | | |
|---|---|---|---|---|---|---|---|---|---|---|---|---|
| | ACC (%) ↑ | TPR (%) ↑ | FPR (%) ↓ | CLIP-Image ↑ | CLIP-Prompt ↑ | FID ↓ | ACC (%) ↑ | TPR (%) ↑ | FPR (%) ↓ | CLIP-Image ↑ | CLIP-Prompt ↑ | FID ↓ |
| EditGuard | 99.69 | 100.0 | 0.62 | **0.98** | - | - | 99.67 | 100.0 | 0.66 | 0.97 | - | - |
| Stable Signature | 99.51 | 99.22 | 0.20 | **0.98** | 0.26 | 27.22 | 99.51 | 99.22 | 0.20 | **0.98** | 0.25 | 20.16 |
| Tree-Ring | 99.72 | **100.0** | 0.56 | 0.86 | 0.25 | 28.37 | 99.96 | 99.98 | 0.06 | 0.82 | **0.25** | 23.02 |
| Gaussian Shading | **100.0** | 100.0 | **0.00** | 0.83 | 0.25 | 29.47 | **100.0** | 100.0 | **0.00** | 0.75 | 0.24 | 23.72 |
| **PAI (Uncondition)** | - | - | - | - | - | - | 98.67 | 97.57 | 0.231 | 0.56 | - | - |
| **PAI (T2I)** | **100.0** | **100.0** | **0.00** | 0.84 | **0.26** | 29.27 | **100.0** | **100.0** | **0.00** | 0.76 | **0.25** | 21.37 |

Figure 7: Evaluation of common image degradations on ownership verification accuracy.

verification accuracy, with T2I reaching 100% TPR. For fidelity, the T2I variant preserves high visual quality and semantic alignment. To further evaluate watermark imperceptibility and user-perceived quality, we conducted a user study (see Appendix A.6.2).

## 6.3 ROBUSTNESS

We focus our robustness evaluation on the T2I setting, as this better reflects a real-world scenario.

**Image Degradation.** As shown in Fig. 7, our method and other inherent schemes remain robust under JPEG compression, noise, blur, and brightness changes, whereas embedded baselines show sharp drops when facing unseen degradations (e.g., EditGuard fails under blur). This confirms that embedded methods largely depend on adversarial or augmentation-based training to achieve robustness against degradation, while inherent watermarking offers a more promising paradigm for real-world protection.

Table 2: Evaluation of watermarking methods under watermark removal and spoofing attacks. We report both attack detection and ownership verification performance, alongside image quality under adversarial conditions.

(a) Watermark removal attacks

| Method | Diffusion Purification Attack | | | Classifier-based Adversarial Attack | | | Imprinting Removal Attack | | | Average | | |
|---|---|---|---|---|---|---|---|---|---|---|---|---|
| | Attack ACC (%) ↑ | Ownership ACC (%) ↑ | PSNR ↑ | Attack ACC (%) ↑ | Ownership ACC (%) ↑ | PSNR ↑ | Attack ACC (%) ↑ | Ownership ACC (%) ↑ | PSNR ↑ | Attack ACC (%) ↑ | Ownership ACC (%) ↑ | PSNR ↑ |
| EditGuard | - | 0.30 | 21.96 | - | **100.0** | 30.30 | - | - | - | - | 50.15 | 26.13 |
| Stable Signature | - | 0.28 | 24.76 | - | 87.62 | **33.00** | - | - | - | - | 43.95 | **28.88** |
| Tree-Ring | - | 67.16 | 24.64 | - | 5.94 | 32.12 | - | 86.00 | **21.92** | - | 53.03 | 26.23 |
| Gaussian Shading | - | **99.98** | 23.92 | - | 99.80 | 31.56 | - | 100.0 | 21.69 | - | **99.93** | 25.72 |
| **PAI (T2I)** | **99.00** | 99.00 | **24.77** | **97.00** | 98.00 | 31.80 | **100.0** | **100.0** | 21.59 | **98.67** | 99.00 | 26.05 |

(b) Spoofing attacks

| Method | Pattern Extraction Attack | | | Reprompting Attack | | | Imprinting Forgery Attack | | | Average | | |
|---|---|---|---|---|---|---|---|---|---|---|---|---|
| | Attack ACC (%) ↑ | Ownership ACC (%) ↑ | PSNR ↑ | Attack ACC (%) ↑ | Ownership ACC (%) ↑ | PSNR ↑ | Attack ACC (%) ↑ | Ownership ACC (%) ↑ | PSNR ↑ | Attack ACC (%) ↑ | Ownership ACC (%) ↑ | PSNR ↑ |
| EditGuard | - | 37.12 | **48.69** | - | - | - | - | - | - | - | 37.12 | **48.69** |
| Stable Signature | - | **99.92** | 40.16 | - | - | - | - | - | - | - | **99.92** | 40.16 |
| Tree-Ring | - | 89.24 | 39.69 | - | 79.36 | - | - | 99.00 | **17.68** | - | 89.20 | 28.69 |
| Gaussian Shading | - | 28.56 | 35.46 | - | 2.74 | - | - | 16.00 | **17.68** | - | 15.77 | 26.57 |
| **PAI (T2I)** | **100.0** | 92.00 | 39.32 | **100.0** | 97.00 | - | **100.0** | **100.0** | 17.32 | **100.0** | 96.33 | 28.32 |

**Watermark Removal Attacks and Spoofing Attacks.** As shown in Tab. 2a and Tab. 2b, existing methods exhibit a clear trade-off: Stable Signature are robust against spoofing but fragile to removal, while Gaussian Shading resists removal but fails under spoofing. This arises from using one-dimensional verification signals (e.g., scalar thresholds or decoded messages), which are unable to capture the differences in attack behaviours. In contrast, our PAI models the directional differences in latent subspace for distinguishing different attack behaviors (Sec. 5.2), achieving 99% O-ACC under removal and 96.3% under spoofing. We further report ownership verification accuracy across different PSNR levels, as shown in Fig. 11.

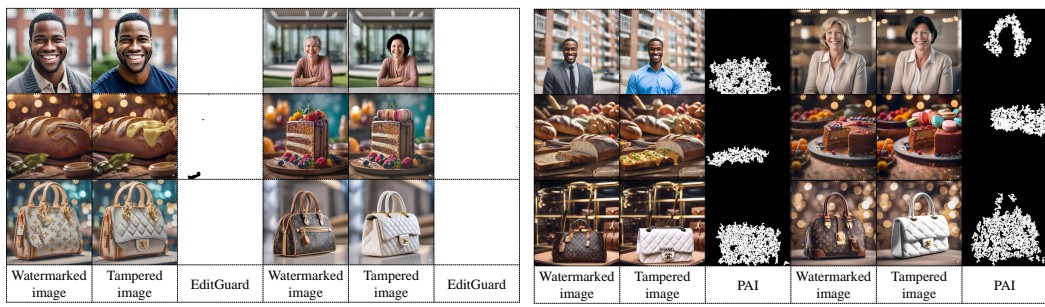

(a) EditGuard localization result      (b) Our PAI localization result

Figure 8: Localization performance comparisons of our PAI and EditGuard on Google's Gemini 2.0 Flash generation model (Google, 2024).

## 6.4 TAMPER LOCALIZATION

Tab. 3 shows that PAI achieves a high level of attack detection and consistently 100% ownership verification across three tampering types: partial edits (PS, Simswap) and full-image semantic editing (Stable Inpainting). Unlike partial edits that preserve much of the original image, full-image methods rewrite the entire content, where EditGuard collapses

Table 3: Comparison of watermarking methods with respect to tampering detection and localization performance. 'A-ACC' and 'O-ACC' denote the accuracy of attack detection and ownership verification, respectively.

| Metric | PS | | Simswap | | Stable Inpainting | | Average | |
|---|---|---|---|---|---|---|---|---|
| | EditGuard | PAI | EditGuard | PAI | EditGuard | PAI | EditGuard | PAI |
| A-ACC ↑ | - | **80.00** | - | **85.00** | - | **73.00** | - | **79.33** |
| O-ACC ↑ | 99.00 | **100.0** | 99.15 | **100.0** | 0.08 | **100.0** | 66.08 | **100.0** |
| F1 ↑ | **86.07** | 66.24 | **85.49** | 84.92 | 34.00 | **80.00** | 68.52 | **77.05** |
| AUC ↑ | **89.68** | 88.20 | **87.62** | 87.10 | 50.00 | **94.00** | 75.77 | **89.77** |
| IoU ↑ | **78.48** | 57.52 | **75.25** | 74.06 | 22.00 | **67.00** | 58.58 | **66.19** |

(O-ACC = 0.08%, F1 = 34.00%). In contrast, PAI remains effective without any task-specific training, reaching F1 of 80.00% and IoU of 67.00%. On average, PAI outperforms EditGuard across all metrics and further succeeds under tampering by a commercial AIGC model Gemini 2.0 Flash (Fig. 8), highlighting its robustness to emerging semantic-level editing tools. We also evaluate localization performance under varying tampered region ratios, with detailed results shown in Fig. 12.

## 6.5 ADAPTIVE ATTACK

We evaluate three adaptive attacks. First, to show the timestamp salt is only for diversity, we randomly tampered with the saved timestamps on watermarked images and observed no degradation in ownership verification (remained 100%), confirming verification is anchored to the private key rather than the public timestamp. Second, under a full white-box assumption where the injection model is known, we attempted to learn a candidate key to evade the verification. Fig. 10a shows the optimization fails to converge, and cannot match and recover the target key. Finally, assuming the PCA projection used for robust verification is also white-box, we optimised learnable image perturbations that push samples toward the benign or non-watermarked regions in PCA space. As shown in Tab. 10b, these attacks do not bypass our pipeline: ownership verification remains at 100% and the attack detection accuracy stays high ($\approx$ 96–100%). We provide a detailed analysis of these white-box adaptive attacks in the Appendix A.6.4.

## 7 CONCLUSION

We propose PAI, a plug-and-play inherent watermarking framework for diffusion-based AIGC image copyright protection. By embedding semantic watermarks during both initialization and sampling, PAI achieves high fidelity and strong robustness without training. Through DDIM inversion, we unify ownership verification, attack detection, and semantic-level tampering localization using a single statistical signal, namely initialization bias. Extensive experiments demonstrate superior performance across 12 attack types, including spoofing and semantic tampering, with 98.43% verification accuracy. PAI offers a practical and principled solution for secure, forensic-ready AIGC content provenance.

## 8 REPRODUCIBILTY STATEMENT

We ensure the reproducibility of our PAI framework by providing detailed descriptions of the system model, dual-stage watermark injection, and verification procedures (Sec. 3, Sec. 4, and Sec. 5). Implementation details, including model configurations, hyperparameters, datasets, and evaluation metrics, are reported in Sec. 6, while additional proofs and extended evaluations are provided in the Appendix. To further support replication, we release the core code and experiment scripts in the repository: `https://github.com/QingyuLiu/PAI`.

## 9 ETHICS STATEMENT

This work aims to protect the copyright and provenance of AI-generated images by introducing a secure, provider-side inherent watermarking framework. Our framework is designed solely for copyright protection and provenance verification. User keys remain private, evaluations rely only on public datasets, and no sensitive personal data is used. We highlight that potential misuse for non-consensual tracking lies outside our intended scope; our contribution is to foster responsible AIGC development by strengthening intellectual property protection while respecting user rights.

## ACKNOWLEDGEMENTS

This work is partially supported by the Zhejiang Provincial Natural Science Foundation of China under Grant (No. LD24F020010), the National Natural Science Foundation of China under Grant (No. 62472372, No. 62572426, No. U24B20182), Shanghai Municipal Science and Technology Major Project (2025SHZDZX025G17), and the National Key R&D Program of China (2024YFB4505300).

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

## A    APPENDIX

### A.1    THE USE OF LARGE LANGUAGE MODELS

Large language models were engaged exclusively for text refinement, including writing clarity and presentation. Theoretical analysis, research methodology, experimental design, and result interpretation were all independently completed by the authors without reliance on such tools.

### A.2    INITIALIZATION BIAS

In this section, we provide the derivation process of the initialization bias of non-watermarked and watermarked images, respectively.

**Formulation.** The process of generating a normal (non-watermarked) image is formulated as $x_t \longrightarrow x_0$ where $t \in [0, T]$. The process of generating a watermarked image using key $K$ is formulated as $x_t^{wm} \longrightarrow x_0^{wm}$, as illustrated in Fig. 3. Given this watermarked image $x_0^{wm}$, we denote the DDIM inversion process with the valid key $K^c$ ($K^c = K$) as $\hat{x}_0^c \longrightarrow \hat{x}_t^c$. Correspondingly, the DDIM inversion process using an invalid key $K^w$ ($K^w \neq K^c$) is expressed as $\hat{x}_0^w \longrightarrow \hat{x}_t^w$. During verification, we compute the initial starting point $x_t^{wm}$ through the initialization function $\mathcal{F}$ (detailed in Eq. 2) and subsequently calculate the initialization bias. When employing the valid key $K^c$, the initialization bias is defined as $\delta_t^c = \hat{x}_t^c - x_t^c$, where $x_t^c = x_t^{wm}$. In contrast, when using an invalid

key $K^w$, the initialization bias becomes $\delta_t^w = \hat{x}_t^w - x^w$, with $x^w \neq x_t^{wm}$ in this scenario. Recalling the deflection function $\mathcal{H}$ in Eq. 3, for simplicity, we denote the deflection coefficients for the valid key and invalid key as $M$ and $N$, respectively, i.e., $M = \gamma \cdot K^c + 1$ and $N = \gamma \cdot K^w + 1$.

**Initialization bias of non-watermarked images using standard DDIM inversion.** The DDIM sampling process is:

$$x_{t-1} = \frac{1}{\sqrt{\alpha_t}}x_t + \left(\sqrt{1 - \bar{\alpha}_{t-1}} - \frac{1}{\sqrt{\alpha_t}}\sqrt{1 - \bar{\alpha}_t}\right)\varepsilon_\theta(x_t, t). \tag{8}$$

Therefore, the ideal inversion process is formulated as:

$$x_t = \sqrt{\alpha_t}x_{t-1} + \left(\sqrt{1 - \bar{\alpha}_t} - \sqrt{\alpha_t - \bar{\alpha}_t}\right)\varepsilon_\theta(x_t, t), \tag{9}$$

However, during the standard DDIM inversion process, since $x_t$ does not exist at timestep $t - 1$, we substitute $\varepsilon_\theta(x_t, t)$ with $\varepsilon_\theta(x_{t-1}, t)$. Consequently, the inversion process of non-watermarked images is expressed as:

$$\hat{x}_t = \sqrt{\alpha_t}\hat{x}_{t-1} + \left(\sqrt{1 - \bar{\alpha}_t} - \sqrt{\alpha_t - \bar{\alpha}_t}\right)\varepsilon_\theta(\hat{x}_{t-1}, t). \tag{10}$$

The initialization bias of non-watermarked images using standard DDIM inversion can be expressed as:

$$
\begin{aligned}
\Delta_t &= \hat{x}_t - x_t \\
&= \sqrt{\alpha_t}\hat{x}_{t-1} + \left(\sqrt{1 - \bar{\alpha}_t} - \sqrt{\alpha_t - \bar{\alpha}_t}\right)\varepsilon_\theta(\hat{x}_{t-1}, t) - \sqrt{\alpha_t}x_{t-1} + \left(\sqrt{1 - \bar{\alpha}_t} - \sqrt{\alpha_t - \bar{\alpha}_t}\right)\varepsilon_\theta(x_t, t) \\
&= \left(\sqrt{1 - \bar{\alpha}_t} - \sqrt{\alpha_t - \bar{\alpha}_t}\right) \cdot \left(\epsilon_\theta(\hat{x}_{t-1}, t) - \epsilon_\theta(x_t, t)\right) + \sqrt{\alpha_t}(\hat{x}_{t-1} - x_{t-1}) \\
&= \underbrace{\left(\sqrt{1 - \bar{\alpha}_t} - \sqrt{\alpha_t - \bar{\alpha}_t}\right) \cdot \left(\epsilon_\theta(\hat{x}_{t-1}, t) - \epsilon_\theta(x_t, t)\right)}_{F_t} + \sqrt{\alpha_t} \cdot \Delta_{t-1}
\end{aligned}
\tag{11}
$$

From Eq. 11, we can see that the initialization bias $\Delta_t$ gradually accumulates with $t$ and reaches its maximum value at $t = T$. We can further derive $\Delta_t$ by conducting a case study for time steps $t$ from 0 to 3. At $t = 0$, since $x_0 = \hat{x}_0$, we have $\Delta_0 = 0$. When $t = 1$, substituting $\Delta_0$ into Eq. 11, we derive $\Delta_1 = F_1$. For $t = 2$, substituting $\Delta_1$ into Eq. 11 yields: $\Delta_2 = F_2 + \frac{\sqrt{\alpha_2}}{M}F_1$. And when $t = 3$, we can similarly get $\Delta_3 = F_3 + \sqrt{\alpha_3}F_2 + \sqrt{\alpha_3\alpha_2}F_1$. By iterating this process, we generalize the final $\Delta_t$ using mathematical induction, expressed as:

$$\Delta_t = \hat{x}_t - x_t = \sum_{i=1}^{t}\sqrt{\frac{\bar{\alpha}_t}{\bar{\alpha}_i}}F_i, \tag{12}$$

$$\text{where } F_i = \left(\sqrt{1 - \bar{\alpha}_i} - \sqrt{\alpha_i - \bar{\alpha}_i}\right) \cdot \left(\epsilon_\theta(\hat{x}_{i-1}, i) - \epsilon_\theta(x_i, i)\right).$$

**Initialization Bias of Watermarked Images.** Similarly, the deflection process in our watermark scheme is:

$$x_{t-1}^{wm} = \frac{M}{\sqrt{\alpha_t}}x_t^{wm} + \left(\sqrt{1 - \bar{\alpha}_{t-1}} - \frac{M}{\sqrt{\alpha_t}}\sqrt{1 - \bar{\alpha}_t}\right)\varepsilon_\theta(x_t^{wm}, t). \tag{13}$$

Therefore, the ideal inversion process is formulated as:

$$x_t^{wm} = \frac{\sqrt{\alpha_t}}{M}x_{t-1}^{wm} + \left(\sqrt{1 - \bar{\alpha}_t} - \frac{\sqrt{\alpha_t - \bar{\alpha}_t}}{M}\right)\varepsilon_\theta(x_t^{wm}, t), \tag{14}$$

where $M = \gamma \cdot K^c + 1$. However, during the DDIM inversion process, since $x_t^{wm}$ does not exist at timestep $t - 1$, we substitute $\varepsilon_\theta(x_t^{wm}, t)$ with $\varepsilon_\theta(x_{t-1}^{wm}, t)$. Consequently, the inversion process of our verification for the valid key $K^c$ is expressed as:

$$\hat{x}_t^c = \frac{\sqrt{\alpha_t}}{M}\hat{x}_{t-1}^c + \left(\sqrt{1 - \bar{\alpha}_t} - \frac{\sqrt{\alpha_t - \bar{\alpha}_t}}{M}\right)\varepsilon_\theta(\hat{x}_{t-1}^c, t). \tag{15}$$

Similarly, for the invalid key $K^w$ (where $K^w \neq K^c$), the verification process becomes:

$$\hat{x}_t^w = \frac{\sqrt{\alpha_t}}{N}\hat{x}_{t-1}^w + \left(\sqrt{1 - \bar{\alpha}_t} - \frac{\sqrt{\alpha_t - \bar{\alpha}_t}}{N}\right)\varepsilon_\theta(\hat{x}_{t-1}^w, t), \tag{16}$$

where $N = \gamma \cdot K^w + 1$.

First, the initialization bias of the valid key can be expressed as:

$$
\begin{aligned}
\delta_t^c &= \hat{x}_t^c - x_t^{wm} \\
&= \frac{\sqrt{\alpha_t}}{M}\hat{x}_{t-1}^c + \left(\sqrt{1-\bar{\alpha}_t} - \frac{\sqrt{\alpha_t - \bar{\alpha}_t}}{M}\right)\varepsilon_\theta\left(\hat{x}_{t-1}^c, t\right) - \frac{\sqrt{\alpha_t}}{M}x_{t-1}^{wm} + \left(\sqrt{1-\bar{\alpha}_t} - \frac{\sqrt{\alpha_t - \bar{\alpha}_t}}{M}\right)\varepsilon_\theta\left(x_t^{wm}, t\right) \\
&= \left(\sqrt{1-\bar{\alpha}_t} - \frac{\sqrt{\alpha_t - \bar{\alpha}_t}}{M}\right)\cdot\left(\epsilon_\theta\left(\hat{x}_{t-1}^c, t\right) - \epsilon_\theta\left(x_t^{wm}, t\right)\right) + \frac{\sqrt{\alpha_t}}{M}\left(\hat{x}_{t-1}^c - x_{t-1}^{wm}\right) \\
&= \underbrace{\left(\sqrt{1-\bar{\alpha}_t} - \frac{\sqrt{\alpha_t - \bar{\alpha}_t}}{M}\right)\cdot\left(\epsilon_\theta\left(\hat{x}_{t-1}^c, t\right) - \epsilon_\theta\left(x_t^{wm}, t\right)\right)}_{F_t^c} + \frac{\sqrt{\alpha_t}}{M}\cdot\delta_{t-1}^c
\end{aligned}
$$

$$(17)$$

From Eq. 17, we can see that the initialization bias $\delta_t^c$ gradually accumulates with $t$ and reaches its maximum value at $t = T$. We can further derive $\delta_t^c$ by conducting a case study for time steps $t$ from 0 to 3. At $t = 0$, since $x_0^{wm} = \hat{x}_0^c$, we have $\delta_0^c = 0$. When $t = 1$, substituting $\delta_0^c$ into Eq. 17, we derive $\delta_1^c = F_1^c$. For $t = 2$, substituting $\delta_1^c$ into Eq. 17 yields: $\delta_2^c = F_2^c + \frac{\sqrt{\alpha_2}}{M}F_1^c$. And when $t = 3$, we can similarly get $\delta_3^c = F_3^c + \frac{\sqrt{\alpha_3}}{M}F_2^c + \frac{\sqrt{\alpha_3\alpha_2}}{M^2}F_1^c$. By iterating this process, we generalize the final $\delta_t^c$ using mathematical induction, expressed as:

$$
\begin{aligned}
\delta_t^c = \hat{x}_t^c - x_t^{wm} &= \sum_{i=1}^{t}\frac{\sqrt{\bar{\alpha}_t}}{\sqrt{\bar{\alpha}_i}M^{t-i}}F_i^c, \\
\text{where } F_i^c &= \left(\sqrt{1-\bar{\alpha}_i} - \frac{\sqrt{\alpha_i - \bar{\alpha}_i}}{M}\right)\cdot\left(\epsilon_\theta\left(\hat{x}_{i-1}^c, i\right) - \epsilon_\theta\left(x_i^{wm}, i\right)\right).
\end{aligned}
$$

$$(18)$$

Regarding the initialization bias of the invalid key, we substitute Eq. 16 to derive:

$$
\delta_t^w = \hat{x}_t^w - x^w = \frac{\sqrt{\alpha_t}}{N}\hat{x}_{t-1}^w + \left(\sqrt{1-\bar{\alpha}_t} - \frac{\sqrt{\alpha_t - \bar{\alpha}_t}}{N}\right)\varepsilon_\theta\left(\hat{x}_{t-1}^w, t\right) - x^w.
$$

$$(19)$$

We can see that $x^w$ solely depends on the invalid key $K^w$ and remains temporally invariant, whereas $\hat{x}_t^w$ evolves dynamically with time step $t$. This temporal dependency motivates a systematic derivation to establish the mathematical relationship between $\hat{x}_t^w$ and the original watermarked trajectory $x_t^{wm}$. To elucidate this relationship, we conduct a temporal case analysis across initial time steps ($t \in \{0, 1, 2, 3\}$). At $t = 0$, since $x_0^{wm} = \hat{x}_0^w$, we obtain $\delta_0^w = \hat{x}_0^w - x^w = x_0^{wm} - x^w$. For $t = 1$, substituting $x_0^{wm} = \hat{x}_0^w$ and the expression of $x_0^{wm}$ from Eq. 13, we derive:

$$
\begin{aligned}
\hat{x}_1^w &= \frac{\sqrt{\alpha_1}}{N}\hat{x}_0^w + \left(\sqrt{1-\bar{\alpha}_1} - \frac{\sqrt{\alpha_1 - \bar{\alpha}_1}}{N}\right)\varepsilon_\theta\left(\hat{x}_0^w, 1\right) \\
&= \frac{\sqrt{\alpha_1}}{N}x_0^{wm} + \left(\sqrt{1-\bar{\alpha}_1} - \frac{\sqrt{\alpha_1 - \bar{\alpha}_1}}{N}\right)\varepsilon_\theta\left(\hat{x}_0^w, 1\right) \qquad \boxed{\leftarrow \text{ substitute } \hat{x}_0^w = x_0^{wm}} \\
&= \frac{\sqrt{\alpha_1}}{N}\left[\frac{M}{\sqrt{\alpha_1}}x_1^{wm} + \left(\sqrt{1-\bar{\alpha}_0} - \frac{M}{\sqrt{\alpha_1}}\sqrt{1-\bar{\alpha}_1}\right)\cdot\varepsilon_\theta\left(x_1^{wm}, 1\right)\right] + \left(\sqrt{1-\bar{\alpha}_1} - \frac{\sqrt{\alpha_1 - \bar{\alpha}_1}}{N}\right)\varepsilon_\theta\left(\hat{x}_0^w, 1\right) \\
&\qquad\qquad\qquad\qquad \boxed{\leftarrow \text{ substitute } x_0^{wm} \text{ in Eq. 13}} \\
&= \frac{M}{N}x_1^{wm} + \left(\sqrt{1-\bar{\alpha}_1} - \frac{\sqrt{\alpha_1 - \bar{\alpha}_1}}{N}\right)\varepsilon_\theta\left(\hat{x}_0^w, 1\right) - \frac{M}{N}\left(\sqrt{1-\bar{\alpha}_1} - \frac{\sqrt{\alpha_1 - \bar{\alpha}_1}}{M}\right)\varepsilon_\theta\left(x_1^{wm}, 1\right).
\end{aligned}
$$

$$(20)$$

Similarly, for $t = 2$, substituting $\hat{x}_1^w$ in Eq. 20 and $x_1^{wm}$ from Eq. 13, we obtain:

$$\hat{x}_2^w = \frac{\sqrt{\alpha_2}}{N}\hat{x}_1^w + \left(\sqrt{1-\bar{\alpha}_2} - \frac{\sqrt{\alpha_2 - \bar{\alpha}_2}}{N}\right)\varepsilon_\theta\left(\hat{x}_1^w, 2\right)$$

$$= \frac{\sqrt{\alpha_2}}{N}\left\{\frac{M}{N}x_1^{wm} + \left(\sqrt{1-\bar{\alpha}_1} - \frac{\sqrt{\alpha_1 - \bar{\alpha}_1}}{N}\right)\varepsilon_\theta\left(\hat{x}_0^w, 1\right) - \frac{M}{N}\left(\sqrt{1-\bar{\alpha}_1} - \frac{\sqrt{\alpha_1 - \bar{\alpha}_1}}{M}\right)\varepsilon_\theta\left(x_1^{wm}, 1\right)\right\} +$$

$$\left(\sqrt{1-\bar{\alpha}_2} - \frac{\sqrt{\alpha_2 - \bar{\alpha}_2}}{N}\right)\cdot\varepsilon_\theta\left(\hat{x}_1^w, 2\right) \qquad \boxed{\leftarrow \text{substitute } \hat{x}_1^w \text{ in Eq. 20}}$$

$$= \frac{\sqrt{\alpha_2}}{N}\left\{\frac{M}{N}\left[\frac{M}{\sqrt{\alpha_2}}x_2^{wm} + \left(\sqrt{1-\bar{\alpha}_1} - \frac{M}{\sqrt{\alpha_2}}\sqrt{1-\bar{\alpha}_2}\right)\varepsilon_\theta\left(x_2^{wm}, 2\right)\right] + \left(\sqrt{1-\bar{\alpha}_1} - \frac{\sqrt{\alpha_1 - \bar{\alpha}_1}}{N}\right)\varepsilon_\theta\left(\hat{x}_0^w, 1\right)\right.$$

$$\left. - \frac{M}{N}\left(\sqrt{1-\bar{\alpha}_1} - \frac{\sqrt{\alpha_1 - \bar{\alpha}_1}}{M}\right)\cdot\varepsilon_\theta\left(x_1^{wm}, 1\right)\right\} + \left(\sqrt{1-\bar{\alpha}_2} - \frac{\sqrt{\alpha_2 - \bar{\alpha}_2}}{N}\right)\varepsilon_\theta\left(\hat{x}_1^w, 2\right)$$

$$\boxed{\leftarrow \text{substitute } x_1^{wm} \text{ in Eq. 13}}$$

$$= \frac{M^2}{N^2}x_2^{wm} + \frac{\sqrt{\alpha_2}}{N}\left[\left(\sqrt{1-\bar{\alpha}_1} - \frac{\sqrt{\alpha_1 - \bar{\alpha}_1}}{N}\right)\varepsilon_\theta\left(\hat{x}_0^w, 1\right) - \frac{M}{N}\left(\sqrt{1-\bar{\alpha}_1} - \frac{\sqrt{\alpha_1 - \bar{\alpha}_1}}{M}\right)\varepsilon_\theta\left(x_1^{wm}, 1\right)\right] +$$

$$\left(\sqrt{1-\bar{\alpha}_2} - \frac{\sqrt{\alpha_2 - \bar{\alpha}_2}}{N}\right)\cdot\varepsilon_\theta\left(\hat{x}_1^w, 2\right) - \frac{M^2}{N^2}\left(\sqrt{1-\bar{\alpha}_2} - \frac{\sqrt{\alpha_2 - \bar{\alpha}_2}}{M}\right)\varepsilon_\theta\left(x_2^{wm}, 2\right).$$

$$(21)$$

A clear pattern emerges in $\hat{x}_1^w$ and $\hat{x}_2^w$. We define $F_i^w$ as:

$$F_i^w = \left(\sqrt{1-\bar{\alpha}_i} - \frac{\sqrt{\alpha_i - \bar{\alpha}_i}}{N}\right)\varepsilon_\theta\left(\hat{x}_{i-1}^w, i\right) - \frac{M^i}{N^i}\left(\sqrt{1-\bar{\alpha}_i} - \frac{\sqrt{\alpha_i - \bar{\alpha}_i}}{M}\right)\cdot\varepsilon_\theta\left(x_i^{wm}, i\right),$$

$$\text{where } i \in [1, T] \tag{22}$$

Thus, $\hat{x}_1^w$ and $\hat{x}_2^w$ can be reformulated as:

$$\hat{x}_1^w = \frac{M}{N}x_1^{wm} + F_1^w$$

$$\hat{x}_2^w = \frac{M^2}{N^2}x_2^{wm} + \frac{\sqrt{\alpha_2}}{N}F_1^w + F_2^w \tag{23}$$

Following this pattern, we derive $\hat{x}_3^w$ in a similar way:

$$\hat{x}_3^w = \frac{\sqrt{\alpha_3}}{N}\hat{x}_2^w + \left(\sqrt{1-\bar{\alpha}_3} - \frac{\sqrt{\alpha_3 - \bar{\alpha}_3}}{N}\right)\varepsilon_\theta\left(\hat{x}_2^w, 3\right)$$

$$= \frac{\sqrt{\alpha_3}}{N}\left\{\frac{M^2}{N^2}x_2^{wm} + \frac{\sqrt{\alpha_2}}{N}F_1^w + F_2^w\right\} + \left(\sqrt{1-\bar{\alpha}_3} - \frac{\sqrt{\alpha_3 - \bar{\alpha}_3}}{N}\right)\varepsilon_\theta\left(\hat{x}_2^w, 3\right)\boxed{\leftarrow \text{substitute } \hat{x}_2^w \text{ in Eq. 21}}$$

$$= \frac{\sqrt{\alpha_3}}{N}\left\{\frac{M^2}{N^2}\left[\frac{M}{\sqrt{\alpha_3}}x_3^{wm} + \left(\sqrt{1-\bar{\alpha}_2} - \frac{M}{\sqrt{\alpha_3}}\sqrt{1-\bar{\alpha}_3}\right)\varepsilon_\theta\left(x_3^{wm}, 3\right)\right] + \frac{\sqrt{\alpha_2}}{N}F_1^w + F_2^w\right\} +$$

$$\left(\sqrt{1-\bar{\alpha}_3} - \frac{\sqrt{\alpha_3 - \bar{\alpha}_3}}{N}\right)\cdot \quad \varepsilon_\theta\left(\hat{x}_2^w, 3\right) \qquad \boxed{\leftarrow \text{substitute } x_2^{wm} \text{ in Eq. 13}}$$

$$= \frac{M^3}{N^3}x_3^{wm} + \frac{\sqrt{\alpha_3\alpha_2}}{N^2}F_1^w + \frac{\sqrt{\alpha_3}}{N}F_2^w + \left(\sqrt{1-\bar{\alpha}_3} - \frac{\sqrt{\alpha_3 - \bar{\alpha}_3}}{N}\right)\varepsilon_\theta\left(\hat{x}_2^w, 3\right) -$$

$$\frac{M^3}{N^3}\left(\sqrt{1-\bar{\alpha}_3} - \frac{\sqrt{\alpha_3 - \bar{\alpha}_3}}{M}\right)\varepsilon_\theta\left(x_3^{wm}, 3\right)$$

$$= \frac{M^3}{N^3}x_3^{wm} + \frac{\sqrt{\alpha_3\alpha_2}}{N^2}F_1^w + \frac{\sqrt{\alpha_3}}{N}F_2^w + F_3^w$$

$$(24)$$

By iterating this process, we generalize the final $\hat{x}_t^w$ via mathematical induction:

$$\hat{x}_t^w = \frac{M^t}{N^t}x_t^{wm} + \sum_{i=1}^{t}\frac{\sqrt{\bar{\alpha}_t}}{\sqrt{\bar{\alpha}_i}N^{t-i}}\cdot F_i^w. \tag{25}$$

Therefore, $\delta_t^w$ can be expressed as:

$$\delta_t^w = \frac{M^t}{N^t}x_t^{wm} - x^w + \sum_{i=1}^{t}\frac{\sqrt{\bar{\alpha}_t}}{\sqrt{\bar{\alpha}_i}N^{t-i}}\cdot F_i^w$$

$$\text{where } F_i^w = \left(\sqrt{1-\bar{\alpha}_i} - \frac{\sqrt{\alpha_i - \bar{\alpha}_i}}{N}\right)\varepsilon_\theta\left(\hat{x}_{i-1}^w, i\right) - \frac{M^i}{N^i}\left(\sqrt{1-\bar{\alpha}_i} - \frac{\sqrt{\alpha_i - \bar{\alpha}_i}}{M}\right)\varepsilon_\theta\left(x_i^{wm}, i\right) \tag{26}$$

**Initialization bias of non-watermarked images with $K$.** If we use a non-watermarked image to try to pass our verification, that is, use the standard DDIM sampling process to get the non-watermarked image, and use $K^q$ to get the inverted noise through our inverted deflection process, then similar to the above derivation process, we can get the initialization bias $\delta_t^q$ of the non-watermarked image with a random $K^q$ as:

$$\delta_t^q = \frac{1}{Q^t} x_t - x^q + \sum_{i=1}^{t} \frac{\sqrt{\bar{\alpha}_t}}{\sqrt{\bar{\alpha}_i} Q^{t-i}} \cdot F_i^q \tag{27}$$

where $F_i^q = \left( \sqrt{1 - \bar{\alpha}_i} - \frac{\sqrt{\alpha_i - \bar{\alpha}_i}}{Q} \right) \varepsilon_\theta \left( \hat{x}_{i-1}^q, i \right) - \frac{1}{Q^i} \left( \sqrt{1 - \bar{\alpha}_i} - \sqrt{\alpha_i - \bar{\alpha}_i} \right) \varepsilon_\theta \left( x_i, i \right)$

where we define $Q = \gamma K^q + 1$ and $x^q = \mathcal{F}(K^q, S)$.

## A.3 THEOREM PROOF

### A.3.1 VALID-KEY VERIFICATION V.S. INVALID-KEY VERIFICATION

We have derived the final expressions for the initialization bias of both valid and invalid keys. Recalling our watermark verification process described in Sec. 5, where copyright ownership is determined by checking whether the initialization bias falls within an empirical threshold range, we now focus on security guarantees. To ensure system security, we will prove in the following analysis that the initialization bias of the valid key is consistently smaller than that of invalid keys, thereby demonstrating that only the legitimate key can pass the verification. Please note that we use the second-order moment to compare the initialization bias between the valid key and the invalid key.

**Theorem.** For an invalid key $K^w$ asymptotically approaching the valid key $K^c$, the second-order moment of the initialization bias $\delta_t^c$ (associated with $K^c$) remains smaller than that of $\delta_t^w$ (associated with $K^w$):

$$\lim_{K^w \to K^c} \mathbb{E}\left[ |\delta_t^c|^2 \right] < \mathbb{E}\left[ |\delta_t^w|^2 \right] \tag{28}$$

**Proof.** For the initialization bias $\delta_t^c$ of the valid key (see Eq. 18), $\varepsilon_\theta(\cdot)$ represents the noise predicted by the model. Theoretically, the noise added at each step in the diffusion model follows an independent standard normal distribution. Therefore, we assume an optimal diffusion model where the predicted noise $\varepsilon_\theta(\cdot)$ adheres to a standard normal distribution $N(0, 1)$ and is mutually independent. Therefore, according to the second moment theory of normal distribution, we have:

$$\mathbb{E}\left[ |\delta_t^c|^2 \right] = \sum_{i=1}^{t} \frac{2 \bar{\alpha}_t}{\bar{\alpha}_i M^{2(t-i)}} \left( \sqrt{1 - \bar{\alpha}_i} - \frac{\sqrt{\alpha_i - \bar{\alpha}_i}}{M} \right)^2. \tag{29}$$

Similarly, for the initialization bias of the invalid key (see Eq. 26), both $x_t^{wm}$ and $x^w$ are generated by our initialization function $\mathcal{F}$ and follow standard normal distributions: $x_t^{wm} = \mathcal{F}(K^c, S), \quad x^w = \mathcal{F}(K^w, S)$. Thus, the second-order moment of the invalid-key initialization bias is expressed as:

$$\mathbb{E}\left[ |\delta_t^w|^2 \right] = \left( \frac{M}{N} \right)^{2t} + 1 + \sum_{i=1}^{t} \frac{\bar{\alpha}_t}{\bar{\alpha}_i N^{2(t-i)}} \left[ \left( \sqrt{1 - \bar{\alpha}_i} - \frac{\sqrt{\alpha_i - \bar{\alpha}_i}}{N} \right)^2 + \left( \frac{M}{N} \right)^{2i} \left( \sqrt{1 - \bar{\alpha}_i} - \frac{\sqrt{\alpha_i - \bar{\alpha}_i}}{M} \right)^2 \right]. \tag{30}$$

From Eq. 29 and Eq. 30, we observe that the summation terms in $\mathbb{E}\left[ |\delta_t^c|^2 \right]$ and $\mathbb{E}\left[ |\delta_t^w|^2 \right]$ exhibit highly analogous structural configurations, differing primarily in the numerical values of $M$ and $N$. Here, $M = \gamma K^c + 1$ and $N = \gamma K^w + 1$. In our watermarking scheme, where $K \sim \mathcal{N}(0, 1)$ and $\gamma = 0.1$ are predefined parameters, both $M$ and $N$ follow the distribution $\mathcal{N}(1, 0.1^2)$. This indicates that $M$ and $N$ possess extremely small variance and remain tightly concentrated around their mean value of 1. Postulating a worst-case scenario where $N$ asymptotically approaches $M$ (i.e., $N \approx M$), we derive:

$$\lim_{N \to M} \mathbb{E}\left[ |\delta_t^w|^2 \right] \approx 2 + \sum_{i=1}^{t} \frac{2 \bar{\alpha}_t}{\bar{\alpha}_i M^{2(t-i)}} \left( \sqrt{1 - \bar{\alpha}_i} - \frac{\sqrt{\alpha_i - \bar{\alpha}_i}}{M} \right)^2$$

$$\approx 2 + \mathbb{E}\left[ |\delta_t^c|^2 \right] \tag{31}$$

Consequently, in the worst case, we theoretically establish the inequality $\mathbb{E}\left[ |\delta_t^c|^2 \right] < \mathbb{E}\left[ |\delta_t^w|^2 \right]$ in the worst case (Theorem of Eq. 28). This analytical result demonstrates that the initialization bias of the valid key is consistently smaller than that of invalid keys, thereby demonstrating that only the legitimate key can pass the verification.

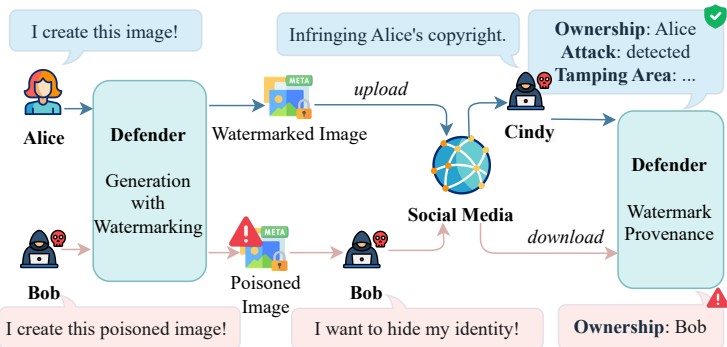

Figure 9: Attack scenarios of our PAI.

### A.3.2 WATERMARKED IMAGES V.S. NON-WATERMARKED IMAGES

We also extend our security analysis to a critical scenario: a strong adversary attempting to bypass watermarking by directly using a clean image, without watermark injection, and applying a random key during inversion. To ensure a fair comparison, we keep the user key and timestamp consistent across both settings, meaning the only difference lies in whether the input image is watermarked. Under this setup, the deflection factors satisfy $M = Q$. Let $\delta_t^q$ denote the initialization bias for the non-watermarked image; its second-order moment is given by:

$$\mathbb{E}\left[|\delta_t^q|^2\right] = \left(\frac{1}{M}\right)^{2t} + 1 + \sum_{i=1}^{t} \frac{\bar{\alpha}_t}{\bar{\alpha}_i M^{2(t-i)}} \left[\left(\sqrt{1-\bar{\alpha}_i} - \frac{\sqrt{\alpha_i - \bar{\alpha}_i}}{M}\right)^2 + \left(\frac{1}{M}\right)^{2i}\left(\sqrt{1-\bar{\alpha}_i} - \sqrt{\alpha_i - \bar{\alpha}_i}\right)^2\right]. \tag{32}$$

Subtracting the second-order moment of the watermarked image's bias $\delta_t^c$, we obtain:

$$\mathbb{E}\left[|\delta_t^q|^2\right] - \mathbb{E}\left[|\delta_t^c|^2\right] = \left(\frac{1}{M}\right)^{2t} + 1 + \sum_{i=1}^{t} \frac{\bar{\alpha}_t}{\bar{\alpha}_i M^{2(t-i)}} \left[\left(\sqrt{1-\bar{\alpha}_i} - \frac{\sqrt{\alpha_i - \bar{\alpha}_i}}{M}\right)^2 + \right.$$

$$\left. \left(\frac{1}{M}\right)^{2i}\left(\sqrt{1-\bar{\alpha}_i} - \sqrt{\alpha_i - \bar{\alpha}_i}\right)^2\right] - \sum_{i=1}^{t} \frac{2\bar{\alpha}_t}{\bar{\alpha}_i M^{2(t-i)}} \left(\sqrt{1-\bar{\alpha}_i} - \frac{\sqrt{\alpha_i - \bar{\alpha}_i}}{M}\right)^2$$

$$= \left(\frac{1}{M}\right)^{2t} + 1 + \sum_{i=1}^{t} \frac{\bar{\alpha}_t}{\bar{\alpha}_i M^{2(t-i)}} \left[\underbrace{\frac{1}{M^{2i}}\left(\sqrt{1-\bar{\alpha}_i} - \sqrt{\alpha_i - \bar{\alpha}_i}\right)^2}_{A^2} - \underbrace{\left(\sqrt{1-\bar{\alpha}_i} - \frac{\sqrt{\alpha_i - \bar{\alpha}_i}}{M}\right)^2}_{B^2}\right] \tag{33}$$

Here, $A^2 - B^2 = (A-B)(A+B)$, and since both $A$ and $B$ are positive, we only need to prove $A - B > 0$ to ensure that the overall difference is positive. By expanding $A - B$, we get:

$$A - B = \frac{1}{M^i}\left(\sqrt{1-\bar{\alpha}_i} - \sqrt{\alpha_i - \bar{\alpha}_i}\right) - \left(\sqrt{1-\bar{\alpha}_i} - \frac{\sqrt{\alpha_i - \bar{\alpha}_i}}{M}\right)$$

$$= \left(\frac{1}{M^i} - 1\right)\sqrt{1-\bar{\alpha}_i} > 0 \tag{34}$$

Since $M \sim \mathcal{N}(1, 0.1^2)$ is tightly concentrated around 1, by Jensen's inequality we know that $\mathbb{E}_M\left[\frac{1}{M^i}\right] > \frac{1}{\mathbb{E}[M]^i} = 1$, which implies $\frac{1}{M^i} > 1$ in expectation. Therefore, $A - B > 0$ holds, and consequently, we have $\mathbb{E}\left[|\delta_t^q|^2\right] > \mathbb{E}\left[|\delta_t^c|^2\right]$.

This analysis demonstrates that even when using the correct key and matching all other variables, non-watermarked images still exhibit statistically larger initialization bias than properly watermarked ones. Thus, the watermark verification process can effectively reject clean images that bypassed the injection phase, confirming the exclusivity of watermark provenance to images generated via the secure watermarking pipeline.

Table 4: Threat model summary of considered attacks. Here, "wm" and "non-wm" denote watermarked and non-watermarked images, respectively.

| Type | Attack | Adversary Capability | Methodology |
|---|---|---|---|
| Degradation | JPEG compression; Noise; Blur; Brightness | Black-box post-processing during dissemination | - |
| Watermark Removal Attack (remove target's watermark) | Diffusion purification attack (Saberi et al., 2023) | Surrogate diffusion model | Reconstruct with a pretrained diffusion model |
| | Classifier-based adversarial attack (Saberi et al., 2023) | Dataset of target's wm/non-wm images | Train a wm/non-wm classifier for the target, then optimize a perturbation to evade the classifier |
| | Imprinting removal attack (Müller et al., 2024) | Surrogate diffusion model | Invert a wm image via a surrogate diffusion model, then optimize a perturbation to maximize the latent distance from the wm image |
| Spoofing Attack (forge target's watermark) | Pattern extraction attack (Yang et al., 2024a) | Dataset of target's wm/non-wm images | Estimate the target watermark as the difference between the average wm vs. non-wm images, then add it to new images |
| | Reprompting attack (Müller et al., 2024) | Surrogate diffusion model | Invert a wm image via surrogate diffusion model, then re-generate with a new prompt while preserving the target watermark |
| | Imprinting forgery attack (Müller et al., 2024) | Surrogate diffusion model | Invert a non-wm image via a surrogate diffusion model, then optimize a perturbation to minimize the latent distance from the wm image |
| Localized Tampering Attack | Sticker addition | Black-box image editing tools | Paste stickers on images |
| | SimSwap (Chen et al., 2020) | Black-box image editing tools | Apply face-swapping to modify identities |
| | Stable Inpainting (Rombach et al., 2022) | Black-box image editing tools | Mask regions and edit via AIGC inpainting |
| Adaptive Attack (full knowledge of our PAI) | Metadata manipulation attack | Black-box model | Alter the timestamp stored in the metadata of wm images; try to evade our detection |
| | Key extraction attack | White-box provider's model | Invert wm images via PAI and optimize a candidate key to minimize initialization bias under the key prior; try to learn keys |
| | PCA space attack | White-box provider's model and PCA model | Invert a wm/non-wm image via PAI and project it into a PCA subspace, then optimize a perturbation to approach the target cluster; try to evade our detection |

## A.4 THREAT MODEL

### A.4.1 IMAGE DEGRADATION

For our degradation robustness evaluation, we define three levels of increasing severity for each degradation type:

- JPEG Compression: Quality factors of 45 (Level 1), 35 (Level 2), and 25 (Level 3).
- Gaussian Noise: Standard deviations of 1 (Level 1), 10 (Level 2), and 50 (Level 3).
- Gaussian Blur: Kernel sizes of 5 (Level 1), 7 (Level 2), and 9 (Level 3).
- Brightness Adjustment: Brightness shifts of –0.1 (Level 1), +0.1 (Level 2), and +0.2 (Level 3).

### A.4.2 ATTACKER SETTING

To comprehensively evaluate watermark robustness, we implement 12 attacks across four categories: watermark removal, spoofing, localized tampering, and adaptive attacks. For watermark removal, spoofing, and adaptive attacks, we use the COCO dataset. For tampering attacks, which involve face-specific manipulations, we use the CelebA-HQ dataset. All attacks are performed on the same set of 5,000 watermarked images generated for a single user. This simulates a strong adversarial setting where the attacker has access to watermarked images of the targeted user, enabling statistical analysis or pattern exploitation tailored to that user. Below, we detail the specific implementation settings for each attack.

- Diffusion Purification Attack. We employ Stable Diffusion v1.5 and add 80 noise steps to reconstruct the 5,000 watermarked images.
- Classifier-Based Adversarial Attack. We train a binary classifier using all 10,000 samples (5,000 watermarked + 5,000 non-watermarked). The classifier is optimized with a learning rate of $1 \times 10^{-4}$ for 100 epochs. To generate adversarial examples, we set adversarial perturbation budget $\varepsilon = \frac{14}{255}$, which corresponds to the strongest setting reported in the original work.
- Imprinting Removal Attack. We use Stable Diffusion XL-base as a surrogate model and train perturbations on 100 watermarked images. To balance attack strength and visual fidelity, we set the training step count to 5 and a learning rate of 0.01.
- Pattern Extraction Attack. We compute the average difference (pattern) between 2,500 watermarked and 2,500 non-watermarked images. This extracted pattern is then added to

a separate set of 2,500 clean images to simulate spoofing. The pattern is applied with a strength factor of 0.1 to preserve perceptual plausibility.

- Reprompting Attack. Using Stable Diffusion XL-base as the surrogate model, we generate 5,000 spoofed images based on COCO prompts. We use 50 DDIM steps and a classifier-free guidance scale of 7.5 during generation to ensure high-quality samples.

- Imprinting Forgery Attack. We adopt Stable Diffusion XL-base as the surrogate model and select 5,000 clean COCO images as cover inputs. 100 adversarial samples are used for training, with a learning rate of 0.01 and 50 optimization steps.

- Copy-and-Paste. We manually curate a library of 31 stickers and emojis from the web. For each of the 5,000 watermarked images, one graphic is randomly selected, resized, and pasted at a random location.

- SimSwap. Following the original SimSwap implementation, we randomly select faces in the 5,000 watermarked images and perform identity swaps using pretrained models.

- Stable Inpainting. We use official Stable Diffusion inpainting tools to edit 5,000 watermarked images. For each image, we randomly mask either the hair or facial expression and apply targeted edits. Hair edits are randomly sampled from six styles: "black", "white", "yellow", "pink", "gray", and "green". Facial expressions are altered among six categories: "smiling", "grinning", "frowning", "pouting", "open-mouthed", and "toothy".

- Metadata Tampering Attack. We generate spoofed timestamps by randomly altering the metadata embedded in the 5,000 watermarked images. This aims to verify that our approach is not time-dependent.

- Key Extraction Attack. Assuming white-box access to the generation model (Stable Diffusion v2.1), we randomly initialize keys and optimize them to minimize watermark verification bias. The generator is frozen, and only the key is updated using MSE loss on initialization bias with a Gaussian regularization penalty. We train for 40 epochs with a learning rate of $1 \times 10^{-4}$.

- PCA Space Attack. Assuming white-box access to the generator (Stable Diffusion v2.1) and the PCA projection used by our verifier, the attacker randomly initializes a perturbation on the latent $z$ of a target image. The goal is to move that image's initialization-bias projection toward a chosen PCA centroid (benign or non-benign). We freeze the generator and a randomly initialized candidate key, since the attacker does not possess the true key. We run this per image for 50 epochs using a pool of 100 watermarked and 100 non-watermarked images. The adversarial loss is:

$$\mathcal{L}(z_{\text{adv}}) = \text{MSE}\big(\delta(z_{\text{adv}}), c\big) + \lambda_{\text{lpips}} \overline{\text{LPIPS}}\big(\mathcal{D}(z_{\text{adv}}), x\big),$$

where $\delta(z_{\text{adv}})$ is the initialization bias of the perturbed latent, $c$ the target centroid in PCA space, and $\mathcal{D}$ the VAE decoder. The MSE term drives the adversarial bias toward the target cluster while the LPIPS term (with $\lambda_{\text{lpips}} = 0.2$) preserves perceptual similarity to the original image $x$. After each step we project $z_{\text{adv}}$ onto the $\ell_{\infty}$-ball $\|z_{\text{adv}} - z\|_{\infty} \leq \epsilon$ with $\epsilon = 0.09$ to bound perturbation magnitude.

## A.5 KEY SPACE

We evaluate the strength of our key design from both theoretical and empirical perspectives.

**Theoretical estimation.** Each user is assigned a private key $K$ with the same shape as the initial noise input of the diffusion model. In Stable Diffusion, this corresponds to a latent tensor of size $4 \times 64 \times 64$, i.e., 16,384 independent components sampled from $\mathcal{N}(0, 1)$. The differential entropy of a standard normal variable is $\frac{1}{2} \log(2\pi e) \approx 2.047$ bits, yielding a total entropy of $16{,}384 \times 2.047 \approx 33{,}548$ bits. This corresponds to an effective key space of $2^{33{,}548}$ elements, forming an extremely high-dimensional, continuous space.

**Empirical estimation.** To estimate the practical limit of distinguishable keys, we evaluated the false positive rate (FPR) of ownership verification under Gaussian perturbations added to $K$ during initialization, as shown in Tab. 5. Using Stable Diffusion ($N = 16{,}384$), we varied the noise ratio to control the mean squared error (MSE) between perturbed and true keys. At a tolerable

Table 5: False positive rates of ownership verification and attack detection under increasing Gaussian noise on user keys.

| Noise Ratio | MSE Distance | Ownership FPR | Attack FPR |
|---|---|---|---|
| 0.30 | 0.18 | 15.27% | 0.64% |
| 0.31 | 0.19 | 10.70% | 0.53% |
| 0.35 | 0.25 | 1.86% | 0.08% |
| 0.36 | 0.26 | 1.05% | 0.08% |
| 0.37 | 0.27 | 0.80% | 0.02% |
| 0.38 | 0.29 | 0.45% | 0.02% |
| 0.39 | 0.30 | 0.00% | 0.00% |

FPR of 0.45%, the decision threshold was $\mathrm{MSE} = 0.29$. The probability that two independent random keys fall within this distance is negligible, estimated via large-deviation analysis as $\mathbb{P}(\mathrm{MSE} < 0.29) \approx 10^{-3828.21}$. Applying the birthday bound (Menezes et al., 1996) with a system-wide collision tolerance of $\delta = 10^{-6}$ gives a maximum user capacity of $U_{\max} \approx 10^{1911.25}$.

## A.6 EVALUATION

### A.6.1 SETTING

**Implementation Details.** We evaluate our watermarking framework in both unconditional and text-to-image (T2I) generation settings. For unconditional generation, we adopt a pre-trained DDPM model (Ho et al., 2020) on CelebA-HQ (Xia et al., 2021). For T2I, we use the publicly available Stable Diffusion v2.1 model (Rombach et al., 2022), generating images conditioned on captions from the COCO (Lin et al., 2015) and CelebA-HQ (Xia et al., 2021) datasets. During generation, we set the number of DDIM sampling steps to 50 with a classifier-free guidance scale of 7.5. The inversion process uses a guidance scale of 1 and an empty prompt, consistent with inversion settings in (Yang et al., 2024b). To inject semantic watermark signals, we apply deflection during the first five sampling steps with a deflection strength of $\gamma = 0.1$. For effectiveness evaluation, each method, including our approach and all baselines, is used to generate 5,000 watermarked images. A shared set of 5,000 non-watermarked images is used across all comparisons. In particular, for post-hoc watermarking methods such as EditGuard (Zhang et al., 2024) and Stable Signature (Fernandez et al., 2023), the watermark is applied for these non-watermarked images to ensure fairness. Robustness evaluation is performed by applying adversarial attacks to the corresponding 5,000 watermarked images for each method. For vanilla verification, we set a threshold ($\tau_{vanilla} = 1.761$) at a significance level of $\alpha = 1 \times 10^{-5}$. For robust verification, we set decision thresholds ($\tau_{robust} = 2.448$) at a significance level of $\alpha = 0.05$. All experiments are implemented in PyTorch with mixed-precision computation and conducted on NVIDIA RTX 3090 GPUs.

**Evaluation Metrics.** We adopt a comprehensive set of metrics to evaluate our watermarking scheme across functionality, image quality, robustness, and forensic capabilities. For binary decision tasks such as ownership verification and attack detection, we report accuracy (ACC), true positive rate (TPR), and false positive rate (FPR). To distinguish between the two tasks, we denote the accuracy of attack detection as Attack ACC (A-ACC) and that of ownership verification as Ownership ACC (O-ACC). Notably, O-ACC consistently reflects the correctness of verifying whether a given image was genuinely generated using the claimed user's key. In the presence of spoofing attacks, where adversaries attempt to forge a watermark and impersonate a legitimate user, a correct verification outcome should reject the forged ownership claim, thereby contributing to a higher O-ACC. Conversely, under watermark removal attacks, although the image may have been manipulated, it was still generated using the correct key and should be recognized as such during verification. Thus, a robust watermarking scheme is expected to achieve high O-ACC under both adversarial scenarios, indicating reliable ownership attribution. To evaluate perceptual quality, we report CLIP-based similarity metrics: CLIP-Prompt measures the semantic alignment between the generated image and its input prompt, while CLIP-Image captures perceptual similarity between the watermarked image and its clean counterpart. Additionally, we report Peak Signal-to-Noise Ratio (PSNR) under adversarial settings to quantify visual distortion. We use it to ensure that attacks remain within realistic threat boundaries, i.e., perturbed images maintain comparable PSNR levels and remain visibly plausible.

For tampering localization, we adopt standard segmentation metrics including F1 score, area under the ROC curve (AUC), and intersection-over-union (IoU), which assess the accuracy and spatial consistency of predicted manipulated regions.

### A.6.2  USER STUDY

We conducted a user study involving 10 participants. Each participant was shown 200 AIGC-generated images, consisting of 50% watermarked and 50% non-watermarked samples, in randomized order. For each image, participants were asked to (1) determine whether a watermark was present, and (2) rate the image quality on a scale of 1 to 5, with higher scores indicating better visual quality. The average accuracy of watermark detection was 49.83%, which is close to random guessing. This result demonstrates that our watermark is visually imperceptible. In terms of image quality, non-watermarked images received an average score of 3.15, while watermarked images received 3.33. The minimal difference suggests that our watermarking process does not degrade perceptual quality.

### A.6.3  ABLATION STUDY

We conduct an ablation study to investigate the impact of the number of deflection steps on watermark effectiveness and robustness. As shown in Tab. 6a, all three configurations (5, 10, and 15 deflection steps) achieve outperforming ownership verification under clean conditions (100.0% ACC, TPR = 100.0, FPR = 0.00), demonstrating that even a small number of deflected steps is sufficient for accurate verification. When evaluating robustness under common image degradations, increasing the number of deflection steps from 5 to 10 or 15 slightly improves stability, particularly under Gaussian noise, where the accuracy improves from 99.00% to 100.0%. Importantly, the CLIP-based similarity metrics remain nearly identical across settings, indicating that increasing deflection steps does not degrade visual or semantic quality.

We also conduct an ablation study to investigate the impact of the deflection intensity $\gamma$ on watermark effectiveness and robustness. As shown in Tab. 6b, the results reveal a clear trade-off between watermark strength and generative fidelity. When $\gamma$ is small (0.1–0.2), PAI achieves perfect clean verification (ACC/TPR = 100%, FPR = 0%) while preserving high perceptual alignment, indicating that mild key-conditioned deflection is sufficient to form a stable watermark signal without disrupting the denoising trajectory. However, when $\gamma$ is increased to 0.3, the perturbation becomes overly strong: both clean verification and degradation robustness degrade sharply (ACC drops to 62.8%), suggesting that excessive semantic deflection distorts the predicted $\hat{x}_0$ and destabilizes the generative path. Based on these observations, we recommend using a conservative deflection strength (e.g., $\gamma = 0.1$) for stable and reliable watermarking in practical deployment.

We further conduct an ablation study to examine how the number of DDIM sampling steps affects watermark verification and robustness. As shown in Tab. 6c, PAI maintains perfect verification performance across all tested configurations (ACC/TPR = 100%, FPR = 0%), indicating that our dual-stage watermark injection does not rely on long sampling trajectories to remain detectable. Increasing the number of steps from 50 to 75 or 100 leads to only marginal variations in CLIP-based perceptual scores, and robustness against common degradations remains uniformly strong (Average = 100% for all settings). These results suggest that the watermark signal—embedded early in the generation process through initialization and deflection—stabilizes quickly and remains consistent regardless of the sampling horizon.

We further analyze how the reference distribution for PCA-based modeling varies with $N$, as shown in Tab. 7. Here $N$ denotes the number of clean (benign) samples used to estimate the benign initialization-bias distribution. Larger $N$ improves the stability of the estimated Gaussian statistics and thus strengthens robustness against removal and spoofing attacks. Results show that ownership verification accuracy remains consistently above 97%, and attack detection accuracy stabilizes at $\sim$ 99%, confirming that our robust verification is insensitive to the number of clean samples $N$.

### A.6.4  WHITE-BOX ADAPTIVE ATTACKS

**Key extraction attack.** We evaluate a white-box adaptive attack in which the adversary, given full access to the generation model (Stable Diffusion v2.1), freezes the generator and attempts to reverse-engineer the target user's private key by optimizing a randomly initialized key to minimize

Table 6: Ablation study on the number of deflection steps, deflection intensity $\gamma$, and DDIM sampling steps. We evaluate watermark verification accuracy under clean conditions and robustness to common image degradations (level 3).

(a) Deflection steps

| Deflection steps | Clean verification | | | | | Image degradation | | | | |
|---|---|---|---|---|---|---|---|---|---|---|
| | ACC | TPR | FPR | CLIP-Image | CLIP-Prompt | Compression | Noise | Blur | Brightness | Average |
| 5 | 100.0 | 100.0 | 0.00 | 0.8499 | 0.2635 | 100.0 | 99.00 | 100.0 | 100.0 | 99.75 |
| 10 | 100.0 | 100.0 | 0.00 | 0.8498 | 0.2628 | 100.0 | 100.0 | 100.0 | 100.0 | 100.0 |
| 15 | 100.0 | 100.0 | 0.00 | 0.8473 | 0.2628 | 100.0 | 100.0 | 100.0 | 100.0 | 100.0 |

(b) Deflection Intensity $\gamma$

| Deflection Intensity $\gamma$ | Clean verification | | | | | Image degradation | | | | |
|---|---|---|---|---|---|---|---|---|---|---|
| | ACC | TPR | FPR | CLIP-Image | CLIP-Prompt | Compression | Noise | Blur | Brightness | Average |
| 0.1 | 100.0 | 100.0 | 0.00 | 0.8499 | 0.2635 | 100.0 | 99.00 | 100.0 | 100.0 | 100.0 |
| 0.2 | 100.0 | 100.0 | 0.00 | 0.8453 | 0.2617 | 100.0 | 100.0 | 100.0 | 100.0 | 100.0 |
| 0.3 | 62.80 | 25.60 | 0.00 | 0.8513 | 0.2623 | 30.00 | 38.60 | 21.00 | 28.60 | 29.55 |

(c) DDIM Step

| DDIM Step | Clean verification | | | | | Image degradation | | | | |
|---|---|---|---|---|---|---|---|---|---|---|
| | ACC | TPR | FPR | CLIP-Image | CLIP-Prompt | Compression | Noise | Blur | Brightness | Average |
| 50 | 100.0 | 100.0 | 0.00 | 0.8499 | 0.2635 | 100.0 | 99.00 | 100.0 | 100.0 | 100.0 |
| 75 | 100.0 | 100.0 | 0.00 | 0.8452 | 0.2622 | 100.0 | 100.0 | 100.0 | 100.0 | 100.0 |
| 100 | 100.0 | 100.0 | 0.00 | 0.8507 | 0.2621 | 100.0 | 100.0 | 100.0 | 100.0 | 100.0 |

Table 7: Impact of the number of clean samples $N$ on robust verification.

| Attack Method | N=75 | | N=100 | | N=125 | | N=150 | |
|---|---|---|---|---|---|---|---|---|
| | Ownership ACC(%)↑ | Attack ACC(%)↑ | Ownership ACC(%)↑ | Attack ACC(%)↑ | Ownership ACC(%)↑ | Attack ACC(%)↑ | Ownership ACC(%)↑ | Attack ACC(%)↑ |
| Diffusion Purification Attack | 99.00 | 99.00 | 99.00 | 99.00 | 99.00 | 99.00 | 99.00 | 99.00 |
| Classifier-based Adversarial Attack | 98.00 | 97.00 | 98.00 | 97.00 | 98.00 | 97.00 | 99.00 | 97.00 |
| Imprinting Removal Attack | 100.0 | 100.0 | 100.0 | 100.0 | 99.00 | 100.0 | 99.00 | 100.0 |
| Pattern Extraction Attack | 93.00 | 100.0 | 92.00 | 100.0 | 92.00 | 100.0 | 91.00 | 100.0 |
| Reprompting Attack | 97.00 | 100.0 | 97.00 | 100.0 | 96.00 | 100.0 | 95.00 | 100.0 |
| Imprinting Forgery Attack | 100.0 | 100.0 | 100.0 | 100.0 | 100.0 | 100.0 | 100.0 | 100.0 |
| **Average** | 97.83 | 99.33 | 97.67 | 99.33 | 97.33 | 99.33 | 97.17 | 99.33 |

the watermark verification bias. The attacker optimizes for 40 epochs with a learning rate of $1 \times 10^{-4}$ using an objective that combines an MSE loss on the initialization bias with Gaussian regularization. As shown in Fig. 10a of the Appendix, the evolution of the attacker's second-order moment of the initialization bias (MSE) remains consistently high throughout training and does not converge to the valid-key baseline (blue dashed line), demonstrating that even with full model access and gradient-based learning the adversary fails to recover a key that passes our verification.

**PCA space attack.** Finally, we implemented a white-box attacker that also has access to the PCA projection used by our robust verification pipeline. The attacker is given 100 watermarked and 100 non-watermarked images from the target user and attempts two goals: (1) *remove* — drive the PCA representation of a watermarked image toward the non-watermarked cluster; and (2) *spoof* — push non-watermarked images toward the benign (watermarked) cluster. For the removal attack, we first mapped the initialization biases of the 100 non-watermarked images into PCA space and used the resulting centroid as the target. Following a PGD-style procedure, we froze all model weights and optimized a small, learnable perturbation on the target watermarked image to minimize its distance to that centroid while constraining perturbation magnitude. The spoofing attack was identical in form but optimized perturbations on non-watermarked images to move their PCA features toward

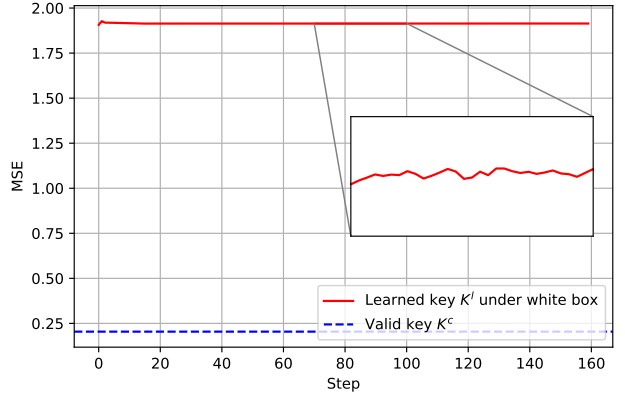

(a) Key extraction attack

| Type | A-ACC | O-ACC |
|---|---|---|
| Removal Attack | 96% | 100.0% |
| Spoofing Attack | 100.0% | 100.0% |

(b) PCA space attack

Figure 10: White-box adaptive attacks.

the benign cluster. As shown in Tab. 10b, even with direct access to the PCA projection and targets' data, the attacker cannot bypass our verification and detection procedures.

### A.6.5 MORE SEMANTIC EDITING ATTACK

To further evaluate robustness under non-diffusion, full-image semantic editing, we compare PAI with EditGuard on InstructPix2Pix editing method (Brooks et al., 2023). As shown in Tab. 8, EditGuard almost completely collapses under this setting (Ownership ACC = 1.80%, F1 = 21.85%, IoU = 13.81%), since its post-hoc embedding can be entirely erased when the editor regenerates the whole image. In contrast, PAI achieves 100% ownership verification and significantly stronger localization performance (F1 = 59.40%, AUC = 85.95%). This demonstrates that our trajectory-level watermark remains detectable even when the entire visual content is semantically rewritten by a non-diffusion generative model.

It is worth noting that the IoU reported for PAI is relatively lower compared to other attacks. The reason is not due to a failure of our localisation mechanism, but due to the absence of a true ground-truth mask in InstructPix2Pix editing. Since these editors do not require an explicit mask for editing, the modified region is unknown. To approximate a reference mask, we follow prior practice and compute a pixel-wise difference between the original and edited images after smoothing and Gaussian denoising. However, this procedure has inherent limitations in capturing semantic-level edits, especially when the editor rewrites texture, lighting, or layout without producing sharp pixel differences. As a result, IoU becomes a naturally conservative metric under this setting, and the "ground-truth" mask itself is only a coarse proxy.

Table 8: Comparison of watermarking methods with respect to tampering detection and localization performance on InstructPix2Pix. 'A-ACC' and 'O-ACC' denote the accuracy of attack detection and ownership verification, respectively.

| Metric | InstructPix2Pix | |
|---|---|---|
| | EditGuard | **PAI** |
| A-ACC ↑ | - | **85.60** |
| O-ACC ↑ | 1.80 | **100.0** |
| F1 ↑ | **21.85** | 59.40 |
| AUC ↑ | **50.01** | 85.95 |
| IoU ↑ | **13.81** | 44.35 |

### A.6.6 RUNTIME ANALYSIS

We report the average runtime (ms) and memory consumption (MB) of each phase in our framework (see Tab. 9), compared to standard DDIM generation and inversion. The results show that generation with PAI introduces only negligible overhead ($< 1\%$) relative to vanilla diffusion sampling. For watermark provenance, vanilla verification and robust verification incur sub-millisecond costs, while tamper localization remains efficient ($\sim 118$ ms). Overall, the total runtime (7950.59 ms) and memory usage (7310 MB) are comparable to standard

Table 9: Runtime analysis.

|  | Time (ms) | Memory Usage (M) |
|---|---|---|
| Standard generation (DDIM sampling) | 7771.78 | 7114 |
| Generation with our PAI | 7827.70 | 7116 |
| Standard inversion (DDIM Inversion) | 7772.17 | 6142 |
| Our Watermark Provenance | | 7310 |
|     Deflection phase | 7830.72 | |
|     Vanilla verification | 0.09 | |
|     Robust verification | 1.25 | |
|     Tamper Localization | 118.53 | |
|     **Total** | 7950.59 | |

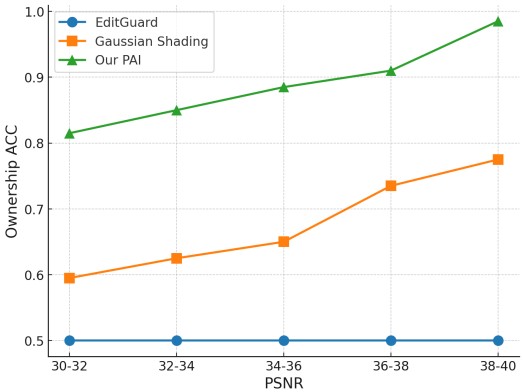

Figure 11: Evaluation of ownership verification accuracy under varying PSNR levels.

inversion, confirming that our design achieves provenance-aware verification with minimal overhead.

### A.6.7 TRANSFERABILITY ANALYSIS

We additionally evaluate the transferability of PAI across different sampling algorithms and model architectures. As shown in Tab. 10, PAI retains perfect verification performance on both DPM-Solver (Lu et al., 2022) and DiT-based diffusion models (Peebles & Xie, 2023), achieving 100% ACC/TPR with zero FPR. For DPM-Solver, the watermark remains fully detectable and robust (Average = 100%), demonstrating that our dual-stage watermark injection does not rely on DDIM-specific dynamics and is compatible with deterministic ODE-based solvers. Similarly, when applied to DiT, a different transformer-based diffusion architecture, PAI still achieves near-perfect robustness under common degradations (Average = 99.75%), confirming that the watermark signal is pre-

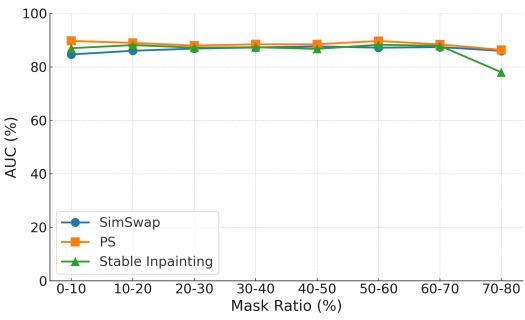

Figure 12: Tampered region analysis under different mask ratios.

Table 10: Transferability of PAI across different sampling algorithms and model architectures. We evaluate watermark verification accuracy under clean conditions and robustness to common image degradations (level 3).

| Sampler and Model | Clean verification | | | | | Image degradation | | | | |
|---|---|---|---|---|---|---|---|---|---|---|
| | ACC | TPR | FPR | CLIP-Image | CLIP-Prompt | Compression | Noise | Blur | Brightness | Average |
| DPM-Solver | 100.0 | 100.0 | 0.00 | 0.8456 | 0.2643 | 100.0 | 100.0 | 100.0 | 100.0 | 100.0 |
| DiT | 100.0 | 100.0 | 0.00 | - | - | 99.40 | 99.60 | 100.0 | 100.0 | 99.75 |

served even when the generative backbone or latent parameterization changes. These results indicate that PAI is inherently sampler-agnostic and architecture-agnostic, as its watermark embedding operates directly on semantic denoising trajectories rather than relying on model-specific features. Thus, PAI can be seamlessly deployed across a wide range of diffusion-style generators without retraining or customization.

### A.6.8 VISUALIZATION RESULT

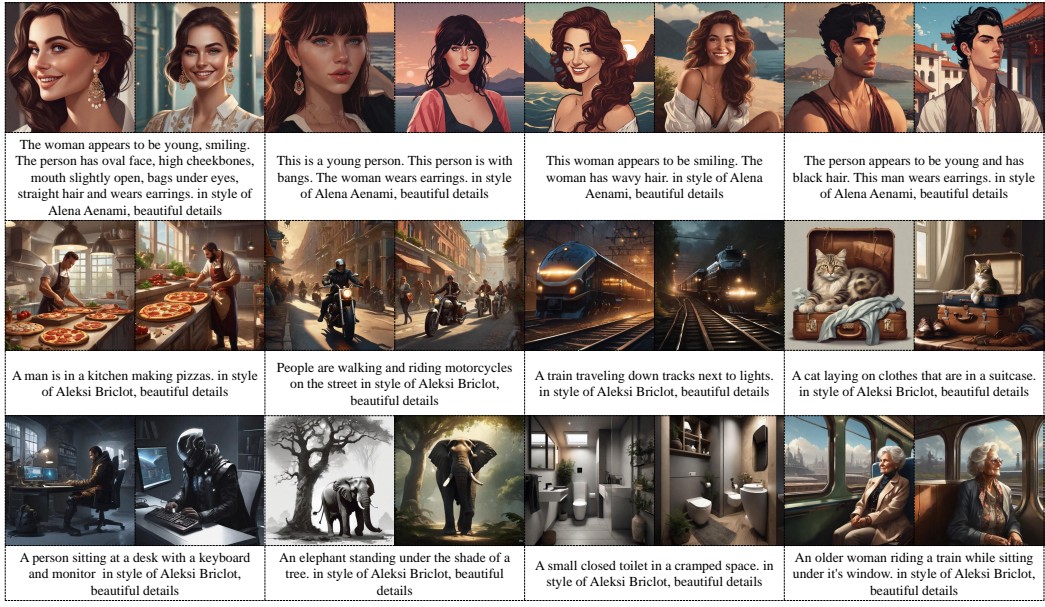

Figure 13: Our watermarked images generated by Stable Diffusion XL. The watermarked images generated using the same key and the same prompt still maintain diversity.

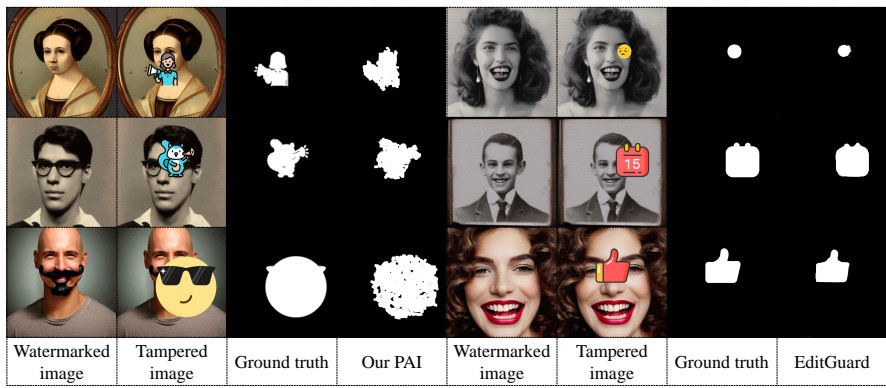

(a) PS-Add Sticker

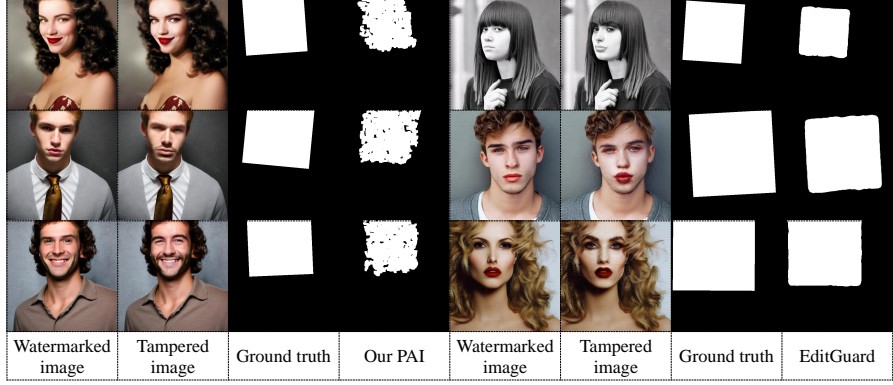

(b) Deepfake-SimSwap

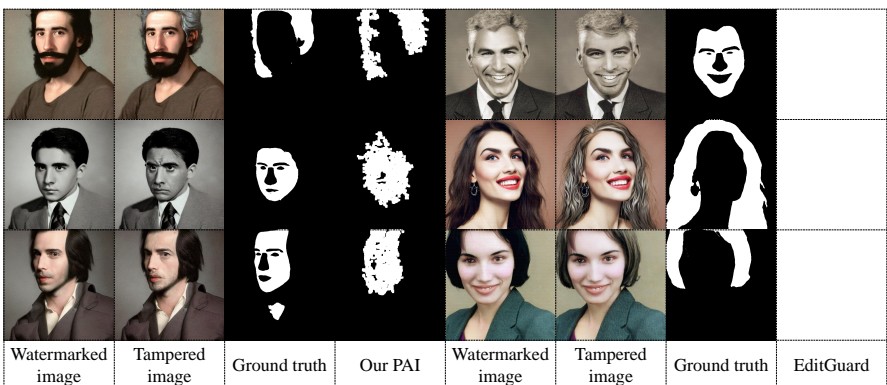

(c) AIGC-Stable Inpainting

Figure 14: Localization performance comparisons of our PAI and EditGuard on PS, Deepfake, and AIGC-based editing methods.

