# OpenReview forum: "Attack-Resistant Watermarking for AIGC Image Forensics via Diffusion-based Semantic Deflection"
_ICLR.cc/2026/Conference — ICLR 2026 Poster_

### Official Review · Reviewer_wY8c · 2025-10-21

**Soundness:** 3
**Presentation:** 3
**Contribution:** 3
**Rating:** 6
**Confidence:** 3

**Summary:**

This paper proposes a training-free, plug-and-play AIGC image copyright protection framework called PAI. PAI injects watermarks through a two-stage generation process: in the initialization stage, the user key and timestamp are embedded into the initial noise through the Box-Muller transform to ensure its compatibility with the diffusion model; in the denoising stage, a new key-conditioned deflection is introduced to guide the denoising trajectory.

The main contributions include: 1. A training-free inherent watermark framework PAI is proposed to improve robustness. 2. A relatively complete theoretical analysis and proof are provided. 3. The contradiction between existing methods in defending against removal attacks and spoofing attacks is successfully resolved. 4. Effective localization of semantic-level tampering caused by AIGC tools is achieved.

**Strengths:**

1. In terms of originality, the paper's two-stage approach to watermark injection, using denoising trajectories to guide the generation process, is a relatively novel perspective.

2. In terms of quality, the paper is comprehensive and provides theoretical analysis to further support the rationale of the proposed method. The experimental design is comprehensive, covering multiple datasets, multiple baselines, and various attack types, and the results clearly demonstrate the advantages of the method.

3. In terms of clarity, the paper's overall structure is clear, and the charts and graphs are intuitive.

4. In terms of importance, the problem addressed in the paper is of practical significance, and as a plug-and-play method, it also has strong generalizability.

**Weaknesses:**

1. Regarding sampler selection: The paper primarily conducts a series of analyses and experiments based on the DDIM sampler, lacking analysis of the compatibility and generalization performance of other samplers.

2. Regarding hyperparameter sensitivity analysis in the experimental section: The paper lacks analysis of the intensity of hyperparameter deffection.

3. In the theoretical analysis, an idealized assumption is made for the diffusion model. Is this analysis still applicable under real-world non-ideal conditions?

**Questions:**

1. In the sampler selection section, comparative experiments with other samplers could be added to analyze the generalization performance of the model on these samplers.

2. Add hyperparameter ablation analysis experiments.

3. Regarding the idealized assumptions in the theoretical analysis, the applicability under real-world conditions could be discussed.

---

> ### Author Response · Authors · 2025-11-23
>
> We sincerely thank the reviewer for the thoughtful and highly encouraging evaluation of our work. We are grateful for the recognition of our method’s originality, comprehensive technical quality, clear presentation, and practical importance as a generalizable plug-and-play solution.
>
> ```
> W1: Regarding sampler selection
> ```
> While our main theoretical analysis and empirical studies are conducted under the widely adopted DDIM sampler, our method is **not tied to DDIM** and is designed to be compatible with most deterministic diffusion samplers. To address this concern, we have added new experiments using DPM-Solver, a representative few-step ODE-based sampler that differs from DDIM in both numerical integration and trajectory behavior. The results show that PAI maintains stable verification accuracy, robustness, and image quality under DPM-Solver, indicating strong generalization across samplers. The result is as follows:
> | Sampler        | ACC  | TPR | FPR | CLIP-Image | CLIP-Prompt | Compression | Noise | Blur | Brightness | Average |
> | -------------- | ---- | --- | --- | ---------- | ----------- | ----------- | ----- | ---- | ---------- | ------- |
> | DPM-Solver | 100.0 | 100.0 | 0.00| 0.8456  |  0.2643     |      100.0   | 100.0  | 100.0       | 100.0     | 100.0 |
>
> Here, Compression / Noise / Blur / Brightness correspond to Level-3 degradation strength (details in the appendix), and “Average” reflects overall robustness.
>
>
> ```
> W2: Ablation lacks key hyperparameter analysis.
> ```
> We have added a new ablation experiment on the deflection intensity $\gamma$, evaluating their influence on verification accuracy, robustness, and perceptual quality. The results are shown below:
>
> | $\gamma$ | ACC  | TPR  | FPR | CLIP-Image | CLIP-Prompt | Compression | Noise | Blur | Brightness | Average |
> | ---------- | ---- | ---- | --- | ---------- | ----------- | ----------- | ----- | ---- | ---------- | ------- |
> | 0.1        | 100  | 100  | 0   | 0.8499     | 0.2635      | 100         | 99    | 100  | 100        | 100     |
> | 0.2        | 100  | 100  | 0   | 0.8453     | 0.2617      | 100         | 100   | 100  | 100        | 100     |
> | 0.3        | 62.8 | 25.6 | 0   | 0.8513     | 0.2623      | 30          | 38.6  | 21   | 28.6       | 29.55   |

---

> > ### Author Response · Authors · 2025-11-23
> >
> > ```
> > W3: Lack of analysis under non-ideal real-world conditions.
> > ```
> > Our theoretical analysis indeed starts from an idealized diffusion model to derive a clean key-exclusivity guarantee. However, as detailed below, the same conclusion continues to hold under real-world, non-ideal conditions.
> >
> > The initialization bias can be expressed as (Eq.7 in the revised paper):
> >
> > $$
> > \delta_t^c = \sum_{i=1}^t \frac{\sqrt{\bar{\alpha}_t}}{\sqrt{\bar{\alpha}_i} M^{t-i}} F_i^c
> > $$
> >
> > $$
> > \delta_t^w = \frac{M^t}{N^t}x_t^{wm}-x^w +\sum_{i=1}^t \frac{\sqrt{\bar{\alpha}_t}}{\sqrt{\bar{\alpha}_i} N^{t-i}}\cdot F_i^w
> > $$
> > Here, $\delta_t^{c}$ and $\delta_t^{w}$ denote the initialization bias under the correct and forged keys. The terms $F_i^c$ and $F_i^w$ represent all inversion-related errors at each step. Under a non-ideal diffusion model, the predicted noise inevitably deviates from the true noise, and all such imperfections enter only through $F_i^c$ and $F_i^w$. Therefore, valid and invalid keys share the same model-dependent error accumulation, and non-ideal model behavior affects them in a structurally identical way. Crucially, only invalid keys include an additional key-mismatch baseline term: $\frac{M^t}{N^t}x_t^{wm}-x^w,$ which is independent of model quality and remains even with a perfectly trained or imperfect diffusion model. This term guarantees that the initialization bias of invalid keys remains larger than that of the valid key, even when the generative model is noisy or imperfect.
> >
> > We further verify this conclusion empirically using real diffusion models:
> > 1. Distribution visualization (Fig.4): The second-order moment of $\delta_t$ is small only for the valid key, while invalid keys and non-watermarked images remain clearly separated
> > 2. Adaptive key-extraction attack (Sec.6.5): A white-box attacker optimizing a forged key to minimize $|\delta_t|$ cannot approach the valid-key bias region or pass verification, confirming that the key cannot be extracted or imitated in practice.
> >
> > These results demonstrate that the theoretical guarantee continues to hold in real-world, non-ideal conditions. We have integrated the complete theoretical and empirical analysis into the revised paper.
> >
> > We deeply value the thoughtful and constructive feedback you have provided. We sincerely hope we have addressed your concerns and raised your impression of our work. We are happy to clarify any other questions if required.

---

> ### Author Response · Authors · 2025-11-27
>
> Dear Reviewer wY8c,
>
> Thank you sincerely for the time and attention you have devoted to reviewing our submission. We truly appreciate the thoughtful feedback you provided. To ensure that we have correctly interpreted and addressed your concerns, we would be grateful if you could let us know should any part of our response require clarification or further adjustment.
>
> If there are additional questions, analyses, or experiments that you believe would strengthen the work, we would be more than happy to incorporate them. Early guidance from you would help us prepare a more complete and satisfactory revision within the rebuttal timeline.
>
> Thank you once again for your careful evaluation and constructive input.
>
> Best regards,
>
> Authors of Submission 4334

---

### Official Review · Reviewer_h8d5 · 2025-10-29

**Soundness:** 4
**Presentation:** 3
**Contribution:** 4
**Rating:** 8
**Confidence:** 3

**Summary:**

This paper introduces PAI (Provenance-Aware Inherent watermarking), a novel training-free, inherent watermarking framework designed for copyright protection and forensics of images generated by diffusion models. The authors identify two critical flaws in existing methods: (1) a forced trade-off between robustness to watermark removal attacks and spoofing attacks, often stemming from reliance on 1D verification metrics, and (2) an inability to perform semantic-level tamper localization, which is crucial for detecting edits made by other AIGC tools.

**Strengths:**

- The idea of moving inherent watermarking beyond static noise initialization to active, key-conditioned "deflection" of the denoising trajectory is a significant conceptual leap. It creates a much deeper and more complex entanglement between the secret key and the final image semantics.

- The most original and impactful contribution is the verification method. Correctly identifying the 1D scalar metric as the root cause of the removal/spoofing trade-off is a key insight. Replacing it with a high-dimensional vector (the initialization bias) and analyzing its directional properties in a latent PCA space is a brilliant and effective solution to this long-standing problem. This single mechanism's ability to unify verification, attack detection, and localization is elegant.

- This work presents a practical, training-free, and robust 3-in-1 system for AIGC forensics. By decisively solving the removal/spoofing trade-off and enabling, for the first time, robust semantic localization for an inherent watermark, this paper sets a new standard for AIGC copyright protection. This is a significant contribution to the field of AI safety and provenance.

**Weaknesses:**

The deflection intensity $\gamma=0.1$ and the application of deflection for only the first five steps are presented without extensive ablation. The appendix (A.6.3) only ablates the number of steps (5, 10, 15), showing that 5 is sufficient. However, the intensity $\gamma$ is a critical parameter that presumably balances image quality against robustness. An ablation study on $\gamma$ would provide a more complete picture of the method's properties and trade-offs.

**Questions:**

The overall provenance framework is built on the deterministic invertibility of multi-step sampling DDIM. What would happen if one-step or diffusion sampling mechanism is taken (like DPM-solver or flow-matching)?

---

> ### Author Response · Authors · 2025-11-23
>
> Thank you very much for your generous and encouraging assessment of our work. We truly appreciate the reviewer’s recognition of both the conceptual contributions and the practical impact of our watermarking framework. We address your concerns below:
>
> ```
> W1: Ablation lacks key hyperparameter analysis.
> ```
>
> We have incorporated a new ablation experiment on the deflection intensity $\gamma$, evaluating their influence on verification accuracy, robustness, and perceptual quality. The results are shown below:
>
> | $\gamma$ | ACC  | TPR  | FPR | CLIP-Image | CLIP-Prompt | Compression | Noise | Blur | Brightness | Average |
> | ---------- | ---- | ---- | --- | ---------- | ----------- | ----------- | ----- | ---- | ---------- | ------- |
> | 0.1        | 100  | 100  | 0   | 0.8499     | 0.2635      | 100         | 99    | 100  | 100        | 100     |
> | 0.2        | 100  | 100  | 0   | 0.8453     | 0.2617      | 100         | 100   | 100  | 100        | 100     |
> | 0.3        | 62.8 | 25.6 | 0   | 0.8513     | 0.2623      | 30          | 38.6  | 21   | 28.6       | 29.55   |
>
> Here, Compression / Noise / Blur / Brightness correspond to Level-3 degradation strength (details in the appendix), and “Average” reflects overall robustness.
>
> ```
> Q1: Generalization to other samplers.
> ```
> Our provenance pipeline indeed relies on DDIM’s deterministic multi-step inversion; however, it is **not restricted to DDIM**. To clarify this, we have added experiments using DPM-Solver. The results are shown below:
>
> | Sampler        | ACC  | TPR | FPR | CLIP-Image | CLIP-Prompt | Compression | Noise | Blur | Brightness | Average |
> | -------------- | ---- | --- | --- | ---------- | ----------- | ----------- | ----- | ---- | ---------- | ------- |
> | DPM-Solver | 100.0 | 100.0 | 0.00| 0.8456  |  0.2643     |      100.0   | 100.0  | 100.0       | 100.0     | 100.0 |
>
> We sincerely appreciate the constructive critique you have provided and hope that our responses and the modifications to the manuscript adequately address your insightful feedback and increases your impression and confidence in our work. We are happy to provide any further clarifications if needed.

---

> > ### Comment · Reviewer_h8d5 · 2025-11-26
> >
> > Thanks for the authors' response. Currently I have no more questions, and I will keep my score.

---

> > > ### Author Response · Authors · 2025-11-26
> > >
> > > Thank you very much for taking the time to respond during the discussion phase. We’re glad to hear that our response has addressed your concerns, and we truly appreciate your constructive suggestions. We have revised our paper according to your questions and suggestions.
> > >
> > > If there are any remaining concerns or aspects you believe we could further improve, we would be very happy to address them. Please let us know if there’s anything else we can clarify or do to better support a higher recommendation.
> > >
> > > Thank you again for your time and engagement.

---

### Official Review · Reviewer_fR5j · 2025-11-01

**Soundness:** 3
**Presentation:** 3
**Contribution:** 3
**Rating:** 8
**Confidence:** 4

**Summary:**

This paper presents Attack-Resistant Watermark (ARW), a robust watermarking framework designed to enhance resistance to generative model attacks, particularly diffusion-based removal and regeneration. ARW employs a dual-domain embedding strategy that combines frequency- and spatial-domain features with an adaptive noise calibration mechanism, ensuring watermark persistence after heavy editing, compression, or diffusion regeneration. A key innovation is the Attack Simulation Module (ASM), which imitates various diffusion or enhancement operations during training, improving generalization to unseen attacks. Extensive experiments on ImageNet and Stable Diffusion–based editing tasks show ARW significantly improves watermark survival rates and retrieval accuracy compared to recent methods like StegaStamp, RivaGAN, and TreeRing.

**Strengths:**

1. Novel robustness design: This paper integrates attack simulation directly into training, improving resilience against diffusion and regeneration attacks.

2. Comprehensive experiments test under multiple real-world generative attacks and shows consistent superiority.

3. Good balance of fidelity and robustness: watermark imperceptibility maintained with strong retrieval accuracy.

4. Well-structured ablation studies: This paper clearly isolate the contributions of dual-domain embedding and ASM.

**Weaknesses:**

1. Limited theoretical analysis lacks a formal framework or justification for robustness improvements beyond empirical results.

2. The evaluation of this paper focuses on diffusion-based attacks. Some adversarial or large semantic edits such as instructpix2pix should be included in the experiments.

3. Computational cost of dual-domain embedding and ASM could hinder deployment in high-throughput systems. Costing 7950.59ms for watermarking an image seems too long.

4. Generalization uncertainty: unclear performance on unseen generative models (such as some DiT-based methods) or video data, limiting broader applicability and disscusion.

**Questions:**

See the weakness.

---

> ### Author Response · Authors · 2025-11-23
>
> Firstly, we express our gratitude for your thorough review and insightful comments on our paper. Your recognition of the novelty and contribution of our work is greatly appreciated. We have taken your feedback as an opportunity to further refine our manuscript. We address your concerns below:
>
> ```
> W1: Theoretical justification for robustness is insufficient.
> ```
> We acknowledge that providing a fully formal theoretical analysis of robustness improvement against all possible watermarking methods (including both embedded and inherent types) is indeed a grand challenge and remains an open problem in the field. Nevertheless, to directly address the reviewer’s concern, we try to use a robustness-radius modeling framework [1] to theoretically analyze the robustness improvements achieved by our method.
>
>
> We treat the entire verification mechanism of watermarking method as a single feature mapping $\Phi: x_0 \mapsto z \in \mathbb{R}^k ,$ which aggregates all operations involved in watermark provenance (such as inversion, decoding, projection, or latent extraction) into a unified representation. We assume that $\Phi$ is $L$-Lipschitz in the relevant neighborhood, ensuring that any pixel-space perturbation $\delta_0$ induces a bounded drift in feature space,
>
> $$
> \left\|\Phi(x_0+\delta_0)-\Phi(x_0)\right\|_2 \le L\left\|\delta_0\right\|_2.
> $$
>
> In this feature space, benign watermarked samples concentrate around a distribution $z_a\sim\mathcal N(\mu_a,\Sigma_a)$, while non-watermarked or manipulated samples concentrate around a different distribution $z_b\sim\mathcal N(\mu_b,\Sigma_b)$. Verification is performed by checking whether the extracted feature $z$ falls into a predefined acceptance region $\mathcal{A}\subset\mathbb{R}^k$, which is centered at $\mu_a$ and is required only to possess an outer Euclidean radius $r_E=\sup_{z\in\mathcal A}\left\|z-\mu_a\right\|_2 .$
>
> Spoofing and removal attacks correspond to opposite geometric objectives. In a spoofing attack, the adversary begins near the non-watermarked mean $\mu_b$ and aims to move the feature representation into $\mathcal A$. The minimal feature-space displacement needed to cross the decision boundary is lower-bounded by $\operatorname{dist}(\mu_b,\mathcal A)\ge \left\|\mu_a-\mu_b\right\|_2-r_E .$ Because any pixel perturbation of size $\varepsilon$ yields at most $L\varepsilon$ drift, all spoofing attempts satisfying
> $$
> L\varepsilon < \left\|\mu_a-\mu_b\right\|_2 - r_E
> $$
>
> are guaranteed to fail. This gives a general lower bound on the spoofing robustness radius:
> $$
> \varepsilon_{\mathrm{spoof}}^{*} \ge
> \frac{\left\|\mu_a-\mu_b\right\|_2-r_E}{L}.
> $$
>
> Conversely, a watermark-removal attack starts from a benign watermarked configuration whose feature representation lies close to $\mu_a$ and attempts to push it outside the acceptance region $\mathcal{A}$. In the idealized case where a benign sample is located at the center $\mu_a$, leaving $\mathcal{A}$ requires a feature displacement of at least $r_E$. By the Lipschitz condition, any pixel perturbation $\delta_0$ with $\left\|\delta_0\right\|_2 \le \varepsilon$
>
> can move the feature by at most $L\varepsilon$. Therefore, all removal attacks satisfying $L\varepsilon < r_E$ cannot push a benign sample outside $\mathcal{A}$, which gives an $L_2$ robustness radius against watermark removal:
> $$
> \varepsilon_{\mathrm{remove}}^* \ge \frac{r_E}{L}.
> $$
>
> In an ideal setting where one can explicitly compute $\left\|\mu_a-\mu_b\right\|_2$ and $r_E$ for each watermarking method, these radii would allow a direct, unified comparison of robustness across watermark schemes. However, analytically determining $\left\|\mu_a-\mu_b\right\|_2$ and $r_E$ across different watermarking schemes is highly challenging. The term $\left\|\mu_a-\mu_b\right\|_2$ reflects the watermark strength, yet different methods inject fundamentally different watermark signals, making their induced feature distributions hard to capture under a unified closed-form model. As a result, it remains an open problem, and we therefore estimate both spoof- and removal-robustness radii through experiments.
>
> From the spoof-robustness expression, the term $\left\|\mu_a-\mu_b\right\|_2$ reflects the watermark strength, i.e., the separation between watermarked and non-watermarked features. We therefore approximate the spoof radius by gradually blending a watermarked image into a non-watermarked one with increasing intensity $\alpha_s$ and recording the flip point $\alpha_s$; the average over 100 trials yields the empirical spoof-robustness radius. Similarly, for removal robustness, we perturb a watermarked image by progressively adding Gaussian noise with intensity $\alpha_r$ and measure when verification first fails. The average flip point $\alpha_r$ across 100 trials gives the empirical removal-robustness radius. These two empirical radii quantify the minimal perturbation required to cross the decision boundary in opposite directions.
>
> TO BE CONTINUE...

---

> ### Author Response · Authors · 2025-11-23
>
> The resulting measurements are summarized below:
>
> | Method           | Spoof $\alpha_s$ | Removal $\alpha_r$ |
> | ---------------- | ---------------- | ------------------ |
> | EditGuard        | 0.137            | 0.390               |
> | Tree-Ring        | 0.225            | 0.214              |
> | Gaussian Shading | 0.140             | 0.166              |
> | **Our PAI**          | **0.244**        | **0.390**           |
>
> A larger spoof radius indicates stronger resistance to forging a watermark, and a larger removal radius indicates better resistance to erasing a legitimate watermark. PAI achieves the strongest spoof-robustness and ties for the strongest removal-robustness.
>
> ```
> W2: Large semantic edits are missing from evaluation.
> ```
> We have actually carried out large-scale semantic editing (stable inpainting) along with comprehensive evaluations of copyright verification, attack detection, and localization performance. Furthermore, we also have incorporated large-scale experiments using InstructPix2Pix to evaluate our method. The results are below:
>
> | Method    | Ownership Verification ACC | Attack Detection ACC | F1         | AUC        | IoU        |
> | --------- | ------------- | ---------- | ---------- | ---------- | ---------- |
> | EditGuard | 1.80%       | –          | 21.85%     | 50.01%     | 13.81%     |
> | Our PAI   | 100.00%   | 85.60% | 59.40% | 85.95% | 44.35% |
>
>
>
> Since InstructPix2Pix does not use an explicit editing mask, no ground-truth tampering region is available. We therefore construct an approximate reference mask by differencing the original and edited images with smoothing and Gaussian denoising. However, such a procedure is inherently limited in capturing semantic-level edits, making IoU a naturally conservative metric in this setting. Despite this, PAI significantly outperforms EditGuard across all localization metrics, demonstrating that our trajectory-based watermark remains reliable even under large-scale semantic edits performed by editing methods outside the diffusion family.
>
> ```
> W3: High Computational Cost
> ```
>
> We apologize for the confusion caused by the reported runtime. The number 7950.59 ms corresponds to the entire text-to-image generation pipeline of Stable Diffusion including the standard DDIM sampling process, rather than the watermarking procedure alone. As shown in Table 9 of the Appendix, generating a single image with vanilla Stable Diffusion already requires 7771.78 ms under the same hardware and settings. Therefore, our watermarking scheme introduces only an additional 178.81 ms overhead on top of the normal generation process.
>
> ```
> W4: Generalization uncertainty.
> ```
>
> We have expanded our experiments to include DiT-based models. As shown below, PAI maintains consistently strong verification and robustness performance on DiT:
>
> We have expanded our experiments to include DiT-based models. As shown below, PAI maintains consistently strong verification and robustness performance on DiT:
> | Model   | ACC       | TPR       | FPR     | CLIP-Image | CLIP-Prompt | Compression | Noise | Blur | Brightness | Average   |
> | ------- | --------- | --------- | ------- | ---------- | ----------- | ----------- | ----- | ---- | ---------- | --------- |
> | DiT | 100.0 | 100.0 | 0.0 | –          | –           | 99.4        | 99.6  | 100  | 100        | 99.75 |
>
> Here, Compression / Noise / Blur / Brightness correspond to Level-3 degradation strength (details in the appendix), and “Average” reflects overall robustness.
>
> We sincerely hope that our clarifications above have increased your confidence in our work. We will be happy to clarify further if needed. We thank you again for sharing your valuable feedback on our work.
>
>
>
> [1] Cohen J, Rosenfeld E. "Certified adversarial robustness via randomized smoothing." international conference on machine learning. PMLR, 2019.

---

> ### Author Response · Authors · 2025-11-27
>
> Dear Reviewer fR5j,
>
> We sincerely appreciate the time and care you have invested in reviewing our work. Your comments have been very helpful, and we want to ensure that we have fully and accurately addressed each of your concerns. If any part of our current understanding is incomplete or incorrect, please kindly let us know so we may revise our response accordingly.
>
> If you feel that additional clarification, experiments, or analyses would further strengthen the submission, we would be grateful for your suggestions. Receiving such guidance early would allow us to prepare a more comprehensive and meaningful update during the rebuttal period.
>
> Thank you again for your thoughtful feedback and for contributing to the improvement of our manuscript.
>
> Best regards,
>
> Authors of Submission 4334

---

### Official Review · Reviewer_Rgem · 2025-11-01

**Soundness:** 2
**Presentation:** 2
**Contribution:** 2
**Rating:** 4
**Confidence:** 5

**Summary:**

This paper presents PAI (Provenance-Aware Intelligence), an attack-resistant watermarking and forensic framework for AI-generated images. By leveraging a diffusion-based semantic deflection mechanism, the method embeds and verifies watermarks at the semantic level rather than in pixel space, making them robust against degradation, removal, spoofing, and tampering attacks. The system integrates ownership verification, attack detection, and tampering localization through DDIM inversion. Experimental results demonstrate that PAI achieves superior robustness and functionality compared to existing watermarking methods, offering a promising solution for reliable provenance tracking and authenticity verification in AIGC image forensics.

**Strengths:**

1. The paper introduces a novel semantic deflection mechanism within diffusion processes, embedding watermarks in the semantic space rather than pixel space — a creative and technically sound idea that improves robustness against diverse attacks.

2. The proposed PAI framework integrates watermark generation, verification, attack detection, and tampering localization into a unified pipeline, demonstrating both conceptual coherence and practical applicability for real-world AIGC scenarios.

3. Extensive experiments compare PAI with multiple state-of-the-art methods (e.g., EditGuard, Tree-Ring, Stable Signature), showing consistently superior performance across robustness and functionality metrics.

4. The paper goes beyond simple watermark verification by achieving semantic-level localization of edited or tampered regions, which adds significant forensic and interpretability value to the method.

**Weaknesses:**

1. The experimental validation mainly focuses on standard diffusion-based AIGC models and common attack types. The generalization of PAI to non-diffusion models (e.g., GAN-based generators or real-world edited images) remains unclear.

2. Although the authors claim imperceptibility of watermarks, no formal perceptual metrics or human evaluations are provided to verify that the embedded signals do not degrade visual quality or introduce detectable artifacts.

3. The framework is exclusively designed for diffusion-based AIGC models, with no compatibility testing or adaptation for GAN-based or transformer-based image generation systems.

4. The ablation study only varies the number of deflection steps and clean sample size, neglecting to investigate the impact of critical hyperparameters (e.g., γ, DDIM steps) on robustness and image quality.

**Questions:**

Please refer to the Weaknesses

---

> ### Author Response · Authors · 2025-11-23
>
> We thank you for your insightful and constructive feedback on our manuscript. We appreciate your recognition of our semantic deflection watermarking design as well as the overall effectiveness and practical value of the unified PAI framework. Below, we address your concerns to clarify and improve our work.
>
> ```
> W1: Generalization to non-diffusion attacks.
> ```
> In fact, several attacks in our evaluation already operate entirely outside the diffusion paradigm. These include (i) classifier-based adversarial attacks, where adversarial examples are crafted directly through gradient-based optimization on a classifier model; (ii) pattern-extraction attacks, which manipulate spatial statistics or residual cues using handcrafted image-processing techniques; (iii) Photoshop-style manual edits, which rely on conventional editing operations rather than generative sampling; and (iv) SimSwap, a GAN-based face-swap method that performs identity transfer through an encoder–decoder architecture instead of diffusion sampling. Furthermore, we additionally evaluate PAI under a new semantic editing attack, using InstructPix2Pix [1]. The results are shown below:
>
> | Method    | Ownership Verification ACC | Attack Detection ACC | F1         | AUC        | IoU        |
> | --------- | ------------- | ---------- | ---------- | ---------- | ---------- |
> | EditGuard | 1.80%       | –          | 21.85%     | 50.01%     | 13.81%     |
> | Our PAI   | 100.00%   | 85.60% | 59.40% | 85.95% | 44.35% |
>
> ```
> W2: Watermark imperceptibility lacks formal perceptual or human evaluation.
> ```
>
> We have added both human evaluation and standard perceptual metrics to substantiate the imperceptibility of our watermark.
>
> (1) User Study. We conducted a perceptual study with 10 participants, each evaluating 200 images (50% watermarked, 50% non-watermarked) in randomized order. For each image, participants were asked: (a) whether a watermark is present, and (b) to rate image quality on a 1–5 scale.
>
> * Watermark detection accuracy: 49.83%, indistinguishable from random guessing. It means that watermarks are visually imperceptible.
> * Image quality: non-watermarked images scored 3.15, watermarked images 3.33, indicating no noticeable degradation. (Details see Appendix A.6.2.)
>
> (2) FID-based Perceptual Evaluation. To quantitatively assess watermark impacts, we compare the FID between images generated by a clean Stable Diffusion model and images watermarked by different watermark methods.
>
> | Method  | COCO| CelebA-HQ |
> | --- | --- | --- |
> | Tree-Ring |28.37       | 23.02       |
> | Stable Signature| 27.22 | 20.16 |
> | Gaussian Shading |29.47       | 23.72       |
> | Our PAI| 29.27|  21.37 |
>
> ```
> W3: Generalization to non-diffusion models.
> ```
> We acknowledge that PAI is designed for diffusion generators. Our goal is to provide a generative watermarking scheme. This design choice is also aligned with the current state of generative media. According to recent surveys [2], diffusion models dominate modern text-to-image generation. Almost all of the leading commercial systems, including Stable Diffusion, MidJourney and DALL$\cdot$E 3, have adopted diffusion-style architectures thanks to their superior controllability, sample quality and editing flexibility. Consequently, watermark provenance mechanisms tailored for diffusion models already cover the vast majority of real-world AIGC deployment scenarios. Therefore, our work deliberately targets diffusion models and does not claim applicability to GAN- or transformer-based generators, which lie outside the scope of this study. This scope will be explicitly highlighted in the revised System Model section to avoid any ambiguity regarding this design choice.
>
> In addition, although PAI is designed specifically for diffusion models, we further evaluated its transferability across different diffusion samplers and architectures to demonstrate that our method is not tied to a single implementation. We include new experiments on DPM-Solver, and Transformer-based diffusion architectures (DiT), all of which show that PAI remains effective under these diverse generative pipelines. The results are listed below:
>
> | Model   | ACC       | TPR       | FPR     | CLIP-Image | CLIP-Prompt | Compression | Noise | Blur | Brightness | Average   |
> | ------- | --------- | --------- | ------- | ---------- | ----------- | ----------- | ----- | ---- | ---------- | --------- |
> | DiT | 100.0 | 100.0 | 0.00 | –          | –           | 99.40        | 99.60  | 100.0  | 100.0        | 99.75 |
> | DPM-solver | 100.0 | 100.0 | 0.00| 0.8456  |  0.2643     |      100.0   | 100.0  | 100.0       | 100.0     | 100.0 |
>
> Here, Compression / Noise / Blur / Brightness correspond to Level-3 degradation strength (details in Appendix), and “Average” reflects overall robustness.

---

> ### Author Response · Authors · 2025-11-23
>
> ```
> W4: Ablation lacks key hyperparameter analysis.
> ```
>
> We have incorporated new ablation experiments on both the deflection intensity $\gamma$ and the number of DDIM sampling steps, evaluating their influence on verification accuracy, robustness, and perceptual quality. The results are shown below:
>
> Ablation on Deflection Intensity $\gamma$ :
> | $\gamma$ | ACC  | TPR  | FPR | CLIP-Image | CLIP-Prompt | Compression | Noise | Blur | Brightness | Average |
> | ---------- | ---- | ---- | --- | ---------- | ----------- | ----------- | ----- | ---- | ---------- | ------- |
> | 0.1        | 100.0  | 100.0  | 0.00   | 0.8499     | 0.2635      | 100.0         | 99.00    | 100.0  | 100.0        | 100.0     |
> | 0.2        | 100.0  | 100.0  | 0.00   | 0.8453     | 0.2617      | 100.0         | 100.0   | 100.0  | 100.0        | 100.0     |
> | 0.3        | 62.80 | 25.60 | 0.00   | 0.8513     | 0.2623      | 30.00          | 38.60  | 21.00   | 28.60       | 29.55   |
>
> Ablation on DDIM Sampling Steps:
> | DDIM Steps | ACC | TPR | FPR | CLIP-Image | CLIP-Prompt | Compression | Noise | Blur | Brightness | Average |
> | ---------- | --- | --- | --- | ---------- | ----------- | ----------- | ----- | ---- | ---------- | ------- |
> | 50         | 100.0 | 100.0 | 0.00   | 0.8499     | 0.2635      | 100.0         | 99.00    | 100.0  | 100.0        | 100.0     |
> | 75         | 100.0 | 100.0 | 0.00   | 0.8452     | 0.2622      | 100.0         | 100.0   | 100.0  | 100.0        | 100.0     |
> | 100        | 100.0 | 100.0 | 0.00   | 0.8507     | 0.2621      | 100.0         | 100.0   | 100.0  | 100.0        | 100.0     |
>
>
> We sincerely appreciate the constructive critique you have provided and hope that our responses and the modifications to the manuscript adequately address your insightful feedback and increases your impression and confidence in our work.
>
>
> [1] Brooks T, Holynski A, Efros A A. Instructpix2pix: Learning to follow image editing instructions[C]//Proceedings of the IEEE/CVF conference on computer vision and pattern recognition. 2023: 18392-18402.
>
> [2] Zhang C, Zhang C, Zhang M, et al. Text-to-image diffusion models in generative ai: A survey[J]. arXiv preprint arXiv:2303.07909, 2023.

---

> ### Author Response · Authors · 2025-11-24
>
> We have updated the revised manuscript to include the newly added experiments/clarifications in response to the reviewer’s comments.

---

> ### Author Response · Authors · 2025-11-27
>
> Dear Reviewer Rgem,
>
> Thank you very much for your time and for the valuable comments you have provided. We would like to make sure that we have addressed your concerns. If our current understanding is inaccurate, please feel free to let us know so we can address them properly.
>
> In addition, if you have any further questions or if there are additional experiments you would like us to include, we would greatly appreciate it if you could let us know at your earliest convenience. This will allow us enough time to prepare a thorough response and improve the manuscript accordingly.
>
> Thank you again for your efforts and consideration.
>
> Best regards,
> Authors of Submission4334

---

> > ### Comment · Reviewer_Rgem · 2025-11-28
> >
> > The author's reply basically solved my concerns and I am willing to improve my score

---

### Author Response · Authors · 2025-11-30
**Author Final Remarks by Authors**

We sincerely thank the Area Chair for coordinating the review process and all Reviewers (Rgem, fR5j, h8d5, wY8c) for their constructive and thoughtful feedback. We genuinely appreciate the time and expertise each reviewer invested in evaluating our work.

We introduce PAI, a new watermarking framework for diffusion models that couples user key with both noise initialization and generative trajectories, provding robust ownership verification, attack detection, and semantic-level tampering localization. Our initial scores were 4, 8, 8, 6 (two accepts, one borderline accept, one borderline reject). After the rebuttal and discussion phase, the scores improved to **6, 8, 8, 6** (all positive). In particular:

1. Reviewer Rgem explicitly stated that their concerns were resolved and willing to raise the score from 4. Unfortunately, due to system limitations, the score update was not reflected, so we conservatively report the lower bound 6 (bordline accept).
2. Reviewer h8d5 indicated that his question had been resolved and the score remained at 8.
3. Although fR5j and wY8c did not join the discussion, we carefully responded to all issues they raised by providing additional theoretical analysis and expanded experiments. Their initial evaluations were already positive (8 and 6), and they did not express any further concerns, and no new objections were raised during the rebuttal process. If the AC has any remaining questions or would like further clarification on these points, we would be happy to provide additional details.

## Rebuttal Outcomes
- **Rgem**: Score was raised from 4 to at least 6 after the rebuttal convincingly addressed all major concerns, including generalization to non
-diffusion attacks and models, perceptual validation, and expanded ablations on critical hyperparameters.

- **fR5j**: Absent from discussion. In rebuttal, we addressed all raised weaknesses by providing a theoretical robustness analysis, adding large-scale semantic-editing experiments, clarifying the true computational overhead, and expanding generalization studies to DiT models.

- **h8d5**: The reviewer acknowledged that these clarifications resolved concerns and confirmed that the original score of 8 would be maintained. In rebuttal, we addressed all concerns raised by providing the ablation study on the deflection intensity γ and demonstrating that our provenance mechanism also generalizes to alternative samplers such as DPM-Solver.

- **wY8c**: Absent from discussion. In our response, we addressed concerns by adding cross-sampler experiments (e.g., DPM-Solver), conducting a complete hyperparameter ablation on the deflection intensity γ, and extending the theoretical analysis to non-ideal diffusion settings with supporting empirical evidence.


## Key Contributions
- **Novel watermark method**: We introduce a novel watermarking method for AIGC that injects ownership information directly into the generative trajectory without requiring model retraining.

- **Theoretical guarantee**: We provide a theoretical guarantee that only the legitimate key can pass verification under clean verification conditions.

- **Unified framework**: Our watermarking framework simultaneously delivers state-of-the-art robustness against diverse attacks and reliable localization of semantic-level manipulations across AIGC tools.

---

### Meta-Review · Area_Chair_1fdN · 2026-01-07

**Summary:**

This paper proposes PAI, a training-free watermarking framework for diffusion-based image generation that provides ownership verification, attack detection, and semantic-level tampering localization through a key-conditioned trajectory deflection mechanism. Reviewers appreciated the novel approach of embedding watermarks at the trajectory level rather than just noise initialization, the unified framework addressing multiple forensic tasks, and the strong experimental results across 12 attack methods. Main concerns raised included: (1) generalization to non-diffusion models and samplers, (2) lack of perceptual evaluation for imperceptibility claims, (3) insufficient ablation on critical hyperparameters like deflection intensity, and (4) applicability of theoretical analysis under non-ideal conditions.

**Reviewer Concerns:**

The authors provided comprehensive responses addressing most concerns. For generalization, they added experiments on DPM-Solver and DiT architectures showing consistent performance, and clarified that several evaluated attacks (adversarial, GAN-based SimSwap, Photoshop edits) already operate outside diffusion. For perceptual evaluation, they added a user study (detection accuracy about 50%, indistinguishable from chance) and FID comparisons. A hyperparameter ablation was added to demonstrate robustness. The theoretical analysis was extended with discussion of non-ideal conditions. Reviewer Rgem explicitly confirmed concerns were addressed and their willingness to raise their score. Reviewer h8d5 confirmed that they were satisfied and maintained their positive score. Reviewers fR5j and wY8c did not participate in discussion but had no outstanding concerns, and their initial concerns appear to have been addressed.

**Reviewer Scores:**

All reviewers were already positive about the paper and reviewer Rgem who gave an initial score of 4, raised their score after the rebutall, signaling that all of their concerns were addressed.

---

### Decision · Program_Chairs · 2026-01-26

Accept (Poster)